# Participatory and Spatial Analyses of Environmental Justice Communities’ Concerns about a Proposed Storm Surge and Flood Protection Seawall

**DOI:** 10.3390/ijerph191811192

**Published:** 2022-09-06

**Authors:** Judith Taylor, Norman S. Levine, Ernest Muhammad, Dwayne E. Porter, Annette M. Watson, Paul A. Sandifer

**Affiliations:** 1Masters in Environmental and Sustainability Studies Program, College of Charleston, Charleston, SC 29424, USA; 2Department of Geology and Environmental Geosciences, Lowcountry Hazards Center, College of Charleston, Charleston, SC 29424, USA; 3Lowcountry Alliance for Model Communities, North Charleston, SC 29405, USA; 4Department of Environmental Health Sciences, Arnold School of Public Health, University of South Carolina, Columbia, SC 29208, USA; 5Department of Political Science, College of Charleston, Charleston, SC 29424, USA; 6Center for Coastal Environmental and Human Health, College of Charleston, Charleston, SC 29424, USA

**Keywords:** environmental justice, Charleston, flooding, seawall, interviews, community, participatory, spatial analysis

## Abstract

In response to increasing threats from sea-level rise and storm surge, the City of Charleston, South Carolina, and the US Army Corps of Engineers (USACE) propose constructing a seawall around the Charleston peninsula. The proposed seawall will terminate close to lower wealth, predominantly minority communities. These communities are identified as environmental justice (EJ) communities due to their history of inequitable burdens of industrial and urban pollution and proximity to highways and US Environmental Protection Agency (EPA) designated Superfund sites. The present study documents community concerns and opinions related to the proposed seawall, existing flooding problems, and other issues. The project was guided by knowledge co-production and participant-observation approaches and included interviews with community members, collection of locality-specific data, GIS mapping to visualize key issues, development of an ArcGIS Story Map, and participation in public meetings. Community concerns are reported in the voices of community members and fell into eight major themes: community connections, drainage, impacts of road infrastructure, displacement, increasing vulnerability, sense of exclusion and isolation, mistrust of government, and civic engagement. Community members were significantly engaged in the study and are the owners of the results. As one of the first US East Coast cities pursuing major structural adaptation for flooding, Charleston is likely to become a model for other cities considering waterfront protection measures. We demonstrate the importance of meaningful engagement to ensure that climate adaptation will benefit all, including marginalized communities, and have as few unintended negative consequences as possible. Bringing more people to the table and creating vibrant, long-term partnerships between academic institutions and community-based organizations that include robust links to governmental organizations should be among the first steps in building inclusive, equitable, and climate resilient cities.

## 1. Introduction

Coastal cities worldwide are facing increasing and existential threats from climate change-related flooding [1,2,3,4]. Charleston, South Carolina (SC), located on the US south Atlantic coast at 32.78° W, 79.93° N) (Figure 1), is a vibrant historic coastal city highly vulnerable to sea-level rise-associated flooding and hurricane storm surge. The relative mean sea level around Charleston has risen 1.07 feet (0.3 m) over the last century and is expected to continue rising more rapidly than previously observed [4,5]. Charleston’s population also increased by 20% between 2010 and 2020 and continues to grow, putting more and more people in risky, low-lying areas [6]. In addition, the mean number of high-tide flood events in Charleston has risen dramatically in the last decade, with an average of 50 high tide flood events of 7 ft (2.1 m) above Mean Lower Low Water (MLLW) occurring between 2012 and 2021, over twice as many as experienced between 2002 and 2011, and many times more than in the 1950s and 1960s (Figure 2). The average high tide in Charleston Harbor is approximately 5.5 ft (1.5 m) above MLLW, with tides greater than 7.0 ft (2.1 m) often causing significant flooding that affects homes, businesses, and transportation [7].

Development, deforestation, and climate change have exacerbated flooding from storm surge events, heavy rains, and every-day high tides, putting pressure on low-lying areas and prompting flood mitigation from the City of Charleston, Charleston County, and other players. In the absence of long-term coastal adaptation or retreat plans, Charleston residents face repeated flood damage and rising insurance premiums. In numerous instances, payouts from claims allowed residents to rebuild flood-damaged homes but did not provide them with a way out of their now, sometimes, difficult-to-sell investments [8] or protection from future flooding. The City of Charleston has adopted several initiatives to combat flooding in recent years, including the establishment of the Resiliency and Sustainability Advisory Committee (RSAC). The RSAC advises city council on topics related to resilience and sustainability, development and adoption of a Flooding and Sea Level Rise Strategy, which uses NOAA (National Oceanic and Atmospheric Administration) projections to set and track resilience goals, and the establishment of Charleston Rainproof, which provides funds for rain gardens throughout the city [9]. The city has also carried out several interdepartmental initiatives to alleviate flooding, including projects and plans to increase the number of trees to absorb excess stormwater, manage natural floodplain functions using FEMA (Federal Emergency Management Agency) standards and guidance, elevate historic structures, and for citizens to self-report, to city officials, among others, flooding and property damage [9]. In 2019, the City and the Historic Charleston Foundation hosted a Dutch Dialogues colloquium that initiated an effort to engage international, national, and local experts to assemble and evaluate data and explore options for the City to better manage the increasing risks of flooding for the long term [9,10]. Despite these measures, Charleston continues to experience impacts from floods regularly, with a record-breaking 89 flood events in 2019 alone (approximately 1 every 4 days), and thousands of structures remaining vulnerable to storm surge [11,12]. As the City continues to consider and adopt flood measures to combat rising sea level and storms, it will be important to include concerns an d needs of all its citizens, in particular those in low-wealth and marginalized communities, in order to ensure equity and that some are not helped at the expense of others.

A large-scale project being considered to protect the City for the long term is the construction of a storm surge wall around most of the peninsula city. This proposal is the outcome of a coastal flood-risk assessment dubbed the Charleston Peninsula Study, conducted by the US Army Corps of Engineers (USACE). The study was initiated in 2018 in collaboration with the City of Charleston, the College of Charleston, The Citadel (the military college of SC), and the SC Department of Transportation (SCDOT), among others [13]. Along with other risk assessments, including the Dutch Dialogues and Charleston’s own Flooding and Sea-Level Rise Strategy [9], this study confirmed that the peninsula city is under existential threat from coastal storm surge and sea-level rise due to low elevation (Figure 3), high hurricane risk, and climate change, and that the City is in dire need of substantial mitigation efforts to protect citizens, businesses, properties, and historical assets from flooding. 

A storm surge wall, accompanied by other measures, was determined to be the most cost-effective way to address flooding in Charleston, beating out six other alternatives containing various combinations of structural and non-structural mitigation options. As currently proposed, the wall would be 9 miles (14.5 km) long and 12 feet (3.7 m) tall, including ground elevation, and equipped with storm surge gates to be closed during low tide before a major storm to accommodate water storage [13]. Additionally, non-structural mitigation measures such as floodproofing, elevation, and structure relocations will be considered in parts of the peninsula not protected by the wall. Also, pumping stations will be located throughout the peninsula to remove water that accumulates behind the wall. The proposed seawall will encircle the downtown peninsula and terminate around the entrance to interstate highway I-26 near Heriot Street (Figure 4). In late 2021, USACE added plans for two stretches of living shorelines (i.e., constructed oyster reefs) along the Ashley River side of the proposed wall (blue lines in Figure 4) to reduce the structure’s impact to surrounding marsh [14].

As proposed, the seawall would terminate at the upper end of the peninsula, leaving areas of the city above the planned terminus outside its protection (Figure 5). Although this area, termed the Charleston Neck, is likely to experience some effects of the wall, such effects were not well considered initially [13]. The Charleston Neck is home to several environmental justice (EJ) communities composed predominantly of African Americans (see below), and uncertainties about the wall and its potential effects led them to raise questions about the proposed wall, future flood risk, and adaptation policies [13,16]. The present study was undertaken to address the following question: What, if any, concerns may environmental justice communities that are not included within the proposed sea wall have about how the wall and flooding might affect their communities? We addressed this question via developing an EJ perspective on the proposed seawall that involved engaging Neck communities, documenting their perspectives and concerns in their own voices, utilizing GIS mapping and analysis to visualize and augment information received from community members, and presenting findings to community residents and other interested parties for use in communicating their perspectives as needed. To ensure effectiveness of communication, we deemed it crucial to report community concerns and issues in their own words. 

## 2. Materials and Methods

### 2.1. Study Area

The Neck of the Charleston peninsula lies at the boundary between the cities of Charleston and North Charleston, along the narrow part of the upper peninsula bounded by the Cooper and Ashley Rivers. This area is home to many people of African and Gullah/Geechee descent, as well as several named communities (Figure 5), including Windsor Place, Rosemont, Chicora/Cherokee, Union Heights, Liberty Hill, Accabee, Howard Heights, and Five Mile [18]. Union Heights was formed in the late 1800s after the American Civil War by previously enslaved people from nearby plantations. Expansion in the 1940s and 1950s, augmented by displacement from the lower Charleston peninsula as it has gentrified since the early 2000s, allowed other communities to grow in adjacent areas [17]. The Charleston Neck now falls in the 96th percentile for SC in population of people of color, the 89th percentile in low-income population, the 99th percentile for proximity to a superfund site, the 98th percentile for proximity to hazardous waste, and the 97th percentile in cancer risk determined by the National Air Toxics Assessment (NATA), among other environmental hazards [19]. The eight Neck communities identified here are considered to be “Environmental Justice (EJ) communities”. The US EPA [20] defines environmental justice as “the fair treatment and meaningful involvement of all people regardless of race, color, national origin, or income, with respect to the development, implementation, and enforcement of environmental laws, regulations, and policies”. Environmental justice communities are typified as low-income and low-wealth, predominantly minority, and disproportionately impacted by previous and/or ongoing negative environmental effects of governmental, business and societal policies and actions [21]. Concerns about unfair environmental burdening of disadvantaged communities are not limited to the US, but are shared internationally [22]. In SC, the SC Department of Health and Environmental Control (DHEC), has established an SC EJ Hub for the purpose of identifying and working to improve the conditions of EJ communities in the state. Communities with environmental concerns self-identify as “EJ”. The Neck communities are considered by DHEC to be EJ communities based on self-identification and the documented instances of disproportionately high levels of environmental hazards experienced by the Neck communities over time [23]. 

In addition to pollution and health worries, the Charleston Neck was originally marshland that was drained and filled in the early 1900s, which has contributed to persistent flooding in the communities that now exist there [24]. The area is at a low elevation and interspersed with tidal creeks, which can contribute to flooding from high tides, heavy rains, and storm surge events. The Lowcountry Alliance for Model Communities (LAMC) was formed in 2005 to address these and other concerns affecting the Neck communities and has been a consistent advocate for environmental justice in the area (https://lamcnc.org/ (last accessed on 4 September 2022). LAMC’s work is supported by the Charleston Community Research to Action Board (CCRAB), which provides research and analysis services for LAMC communities. LAMC was a full partner in the present project and LAMC’s Executive Director is a co-author of this paper, in keeping with practices of knowledge coproduction and community-based participatory research. 

USACE’s stated reasoning for the termination of the seawall at the Charleston Neck is that Neck areas typically are at a higher elevation than the rest of the peninsula, and therefore do not require the wall’s protection [25]. While parts of the Charleston Neck are at a higher elevation than the downtown area, there are parts that are below 12 ft (3.7 m) in elevation, which would put them below the height of the wall and outside its protection (Figure 6 and Figure 7). When this study began in early 2021, the USACE had not released its full Environmental Impact Statement (EIS), investigated environmental justice impacts of the wall, or announced intentions to pursue floodproofing and home elevation in Rosemont, the only one of the Neck EJ communities that is included within the City of Charleston. These gaps highlighted the need to gauge residents’ opinions on these topics as they started to evolve between 2021 and 2022, and to ensure they became part of the conversation. Further, the Neck considers itself a part of the overall social-ecological system that is Charleston, and thus the USACE’s proposals for the wall terminus and related mitigatory actions may require additional consideration from cultural, as well as physical, standpoints.

### 2.2. Research Approach

This mixed-methods study encompassed three phases: (1) participatory outreach and trust-building via interviews and participant observation, (2) development of an EJ profile with the use of GIS, and (3) delivery of findings to the Neck communities via an interactive, public meeting. Our work followed a knowledge co-production approach and leveraged participatory research, methodologies that prioritize joint knowledge creation to achieve community-owned outcomes [26,27,28,29]. Participant-observation approaches were selected to ensure that marginalized communities were consulted and engaged throughout the entirety of the study, and that their needs and concerns were represented within the complexity of a social-ecological analysis [30]. Participant-observation is defined as “a method to turn witnessed experiences into qualitative data” and involves researchers observing, recording, and processing field data on elements of participants’ daily lives [26]. Community engagement via participant observation by the first author was facilitated by participation of LAMC in all aspects of the work and by the involvement of four of the authors in an EPA-funded project entitled “EJ Strong: Strengthening Communities for Disaster Risk Reduction, Response and Recovery in South Carolina” that focused predominantly on the Neck communities. The study plan was approved by the College of Charleston’s Institutional Review Board (IRB), prior to the initiation of interviews and other formal contacts with community members (CofC IRB # 2021-05-003). The EPA’s Technical Guidance for Assessing Environmental Justice in Regulatory Analysis (EJTG) document, including its Human Health Risk Assessment Framework [31], guided our development of objectives and community engagement activities.

Phase 1, participant-observation and participatory outreach, focused on building trust with community members, leaders, and officials from USACE, Charleston, and local resilience organizations, obtaining insight into their major concerns related to flooding and the proposed surge wall, and learning from them what information should be included in the study’s spatial component. Additional contextual information was provided by previous work, which described the Neck area and its history [24]. Next, a semi-structured interview plan was prepared, consisting of a list of prepared questions (Table 1) that also allowed for follow-up questions and clarification [32]. Questions explored residents’ connections to their community, concerns about flooding and other issues, community needs, and views on resiliency, among other things. Note that this project did not use a survey methodology, but a community-based qualitative methodology that was triangulated between interviews and participant-observation. The interview question design was based on consultations with community leaders and through the direct participation of the LAMC organization, who ensured the relevance and validity of the qualitative data to collect.

Community members willing to contribute to this study were sampled via a snowball approach and identified through LAMC, word-of-mouth recommendations from community residents, and flyers distributed during community meetings and canvassing. Interviewees were chosen on the basis of their community affiliation, interest, and willingness to be interviewed. No attempt was made to select a “representative” sample of the populations as that was beyond the scope of the project and in accordance with community-based methodologies. All interviews were preceded by a review of an IRB-approved consent form to explain the study clearly, request consent to record audio, and explain how collected data would be kept confidential and stored (see Appendix A). Affirmative consent was received from each person interviewed before an interview was initiated. Although only seven community members were interviewed, the aim of the interviews was to obtain a diversity of community perspectives, not to be statistically generalizable. Eight interviews in total were conducted, including seven of persons characterized as community resident/leaders (Table 2) and one as a city official. The official was excluded from the table to preserve anonymity and because different interview questions were used in that interview (see Table 1). For interviews of community residents, the mean interview duration was 52:26 min (range: 25:49–1:51:26). While the study would have benefitted from additional participants, opportunities to reach others were severely limited by the COVID-19 pandemic. The resulting interview data were supplemented by field observations collected via participant-observation during five community meetings, community canvassing, informal conversations with residents and USACE project officials, and from various email chains between community leaders, residents, and partner organizations (Appendix A). These and other opportunities allowed us to reach additional people who may not have been willing or able to sit for a full interview. 

Information gathered from the semi-structured interviews and field observations between June and December 2021 was incorporated into the GIS analysis and EJ profile generation. This project did not use a survey methodology, but a community-based qualitative methodology that was triangulated between interviews and participant-observation. The interview question design was based on consultations with community leaders and through the direct participation of the LAMC organization, who ensured the relevance and validity of the qualitative data to collect.

Data collected in this study were from semi-structured interviews, not surveys, and participant observation during meetings, workshops, and other events in the communities. These are qualitative data that were not intended for standard statistical analysis for generalizability, because the researchers were searching for all possible community ideas regarding the cause and effects of flooding and plans for mitigation—an inductive rather than deductive research design. In addition, the interview sample size was insufficient to discern potential relationships. Our intent from the outset was to identify important themes via the qualitative coding of interview responses, then to link them with spatial data using GIS, so the statistical relevance of the observed phenomena to map was not necessary. We employed a commonly used software, NVivo [33] to code qualitative responses to interview questions and notes taken during community meetings, following the approach of Cope [34]. Content-based codes were created for each interview question, which included the participant’s birth year, the one-to-three word description of the participant’s community, any observed changes in the community, why the community is important, daily activities, important places, experiences with flooding, locations of flooding, any changes in flooding, perceptions of resilience, perceived benefits of the seawall, concerns related to and questions about the seawall, and suggestions for the study. 

As needed, more codes were added to capture additional information offered by participants during the interviews that did not necessarily relate to the questions posed. The additional information included personal stories, concerns in the communities not related to flooding or the seawall, opinions about the seawall that did not resemble a concern or potential benefit, suggestions for the City of Charleston and USACE, and recommendations for community improvements. One- to three-word descriptors were further separated into words with negative connotations such as “underserved” and “isolated” as well as positively correlated words such as “home” and “important” to better reveal both critical and positive descriptors. Changes in participants’ communities included changes in the types of residents living there, community demographics, amenities, and upkeep. Daily activities included fishing, gardening, and reliance on various modes of transportation. Important places included churches, natural features, convenience stores, and gathering spaces. Personal stories covered family history and involvement with local government and advocacy organizations. Flooding experiences incorporated specific locations of flooding, issues with drainage systems, and impacts of development and roads on flooding and runoff. Opinions about the seawall included potential benefits, concerns, and questions. The suggestions offered by participants were for community improvements, the City of Charleston, and USACE (Table 3). 

Phase 2 involved creating an EJ profile to visualize the eight themes (see Results) identified in phase 1. For example, the study results identified increased flooding as a major concern, so an elevation layer to identify low-lying areas, the flood-risk layer developed during the Dutch Dialogues, and community- identified important places and frequently flooded areas were incorporated to visualize areas of concern. A drone was utilized to capture finer-scaled elevation data for more detailed investigation of flooding in the Rosemont community. Additionally, any place-based or spatial information collected in the interviews and field observations was mapped in ArcGIS Pro, including participant-identified locations of flooding, important places, and locations of industrial facilities and Superfund sites (Appendix A). Maps and data were compiled in an ArcGIS Story Map, which combined visualizations and commentary worded in plain, non-technical language. We anticipated that creating maps informed by community members and key informants would increase legitimacy of results and allow for the integration of qualitative and quantitative data to produce a more nuanced picture of potential impacts from the surge wall [35,36]. Screenshots of the Story Map are included in the Appendix A.

Phase 3 consisted of a report-back to community members and study participants via an interactive meeting held in the Rosemont community. The ArcGIS StoryMap developed in Phase 2 was presented and the link was shared with attendees to access afterwards. Additionally, the meeting was recorded and the recording was made available to those who could not attend in person or virtually and archived. The presentation elicited a vibrant discussion, with community members actively engaging in questions and answers and sharing their thoughts. Specifically, community members were encouraged to furnish additional local knowledge that could be incorporated into the study, evaluate and identify preferred courses of action, and engage with information on how climate change and the seawall may impact their lives [36,37]. Participants were asked to share their responses to the material, how they interpreted the data, any suggestions on things to change, reword, or add, and what they believed they could use the report for in the future.

## 3. Results

### 3.1. Community Concerns and Opinions

Findings from the community interviews and field observations were contextualized with spatial data, literature, and publicly available data from the US Census. Interview data processing started with creation of content-based codes for each question asked during the interviews. From these initial content-based codes, eight groupings or themes emerged: (1) community connections, (2) drainage, (3) impacts of road infrastructure, (4) displacement, (5) increasing vulnerability, (6) sense of exclusion, (7) mistrust of government, and (8) civic engagement. Qualitative data were coded using the eight primary themes and illustrative quotes (in italics) from participant interviews are compiled below by theme.

***Community connections.*** Throughout the research period, in both formal interviews and in community meetings, participants expressed strong connections to their communities and the Charleston Neck. For example, two of the interviewees stated:


*“If these areas didn’t exist, these communities didn’t exist, I wouldn’t be who I am.”*

*“I see it as a badge of honor to be able to be a resident there [in Rosemont], I see it as a badge of honor for me to live there and raise my children there.”*


Participants had strong ties to both their communities and the land, associating their personal identities with their communities and fellow residents and recognizing the cultural importance of the Neck communities as historic communities of color.


*“…there’s a lot of historical values about Rosemont. It’s one of the first black African American neighborhoods established after the Reconstruction Era…the land, the people, are all very important in the history of the City of Charleston.”*


Historical, culturally-sound, important, and African American were among the words participants used to describe their communities (Figure 8), further exemplifying people’s ties to their communities and identities. The Neck has a large African-American population compared to surrounding areas, with 90% of the Charleston Neck population identifying as people of color, versus only 37% on the Charleston peninsula [38].

Importantly, participants identified with the Neck as a social-ecological system, expressing connections to their communities and other residents as well as to the land and natural features unique to the Neck. When asked to describe the locations of important places, participants mentioned natural features such as Shipyard Creek, Northbridge Park, and the marsh between Rosemont and the Ashley River, along with churches and grocery stores among others. Some residents identified fishing as an important activity, and culturally many of the African-American residents are of the Gullah/Geechee culture, who historically fished for subsistence.

***Drainage.*** Residents and leaders from Rosemont, Union Heights, and Garden Hill all reported issues with poor drainage in their communities, including mentions of flooding, standing water, blocked drains, and lack of stormwater infrastructure. Participants indicated that their communities either lacked stormwater infrastructure and drains entirely, or that their communities had these features, but they were not cleaned regularly or maintained properly and, therefore, functioned poorly if at all.


*“When I was a kid in the city, it didn’t do much good, but one of the things they did about once a quarter is they’d open the drains on the side of the street, you know next to the sidewalk…they’d reach into the drains and they’d pull stuff up out of the drains and make sure the drains were clear. That kinda thing doesn’t happen anymore.”*

*“…we don’t even have a drainage system on the sidewalks or our roads…there’s no place for the water to go except for downhill.”*


Participants also mentioned a desire for more green space in their communities for gatherings and to mitigate the impacts of roads, potentially reduce flooding, and improve drainage. The only large green space in Rosemont is the park towards the Northern end of the community, and participants have reported that this park floods frequently (Figure 9).


*“A neighborhood where the people have a place to congregate like a park or something with activities…a clean space, like I said where somewhere people can go to relax.”*

*“I would like to see more green spaces, drainage improvement back here…”*


***Impacts of road infrastructure.*** Roads and transportation infrastructure were consistently identified as contributing to flooding and other issues such as air pollution in the Neck communities. When asked whether participants had observed any changes in the frequency or severity of flooding, those who described flooding as getting worse attributed this to road construction and development.


*“…Rosemont, we know that it floods more frequently now because the water runs off of the interstate.”*

*“...you can see under the bridge, like if you go further [downtown] you see paved areas, but this is what we have [unpaved and large puddles], and you see the people coming through with this big truck over here [a large truck was parked underneath the bridge].”*

*“It would be a lot better if there were just trees and not the intrusive highways and trucks and other traffic that comes with it, and the pollution that they bring with them.”*


The history and legacy of road construction through the Neck stretches back to the 1960s, when Interstate-26 and eventually Interstate-526 were built through these majority black communities, cutting them off from each other and nearby urban centers [40]. In many cases, the Neck communities have been physically shaped by roads, leading to increased exposure to traffic and air pollution, as well as isolation from the cities of Charleston and North Charleston that have persisted to the present (Figure 10 and Figure 11).

Rosemont residents expressed concerns with a sound wall built between the community and the I-26 ramp by the SCDOT to reduce noise and automobile exhaust from entering the community (Figure 12). Many believe the wall has contributed to flooding by replacing ditches that used to stretch alongside the community, while others see the wall as another feature isolating the community from Charleston.


*“…once [SCDOT] added [the sound wall] and rebuilt that road a lot of that stormwater runoff…I believe it’s come down into the neighborhood.”*


Available elevation data for the Rosemont community predated construction of the sound wall by the SCDOT in 2020. To fill this gap, a drone was used to collect updated elevation data at a high resolution to investigate residents’ concerns that the wall filled in ditches that used to lie along the edge of Doscher Avenue, resulting in increased flooding for properties closest to the wall as well as others down-slope. A DJI drone was flown above Rosemont and captured surface elevation in a double grid pattern, meaning the drone rotated 90° to capture data and images at multiple orientations. This type of flight pattern is ideal for capturing surface elevation data [41]. Comparing the previously available digital elevation model (DEM) with the new higher resolution digital surface model (DSM) of Rosemont captured by the drone, the latter clearly shows areas of low elevation between the wall and the houses closest to the wall (Figure 13). It is reasonable to conclude that stormwater that used to flow more freely here is constrained by the wall and pools in low elevation areas, resulting in flooding for these homes and downhill Rosemont during heavy rainfall.

***Displacement.*** A major concern present in interview and field observation results is residents’ fear of displacement from their homes and communities in the Charleston Neck. This fear is related to participants’ strong connections to their communities and their desire to remain in their homes. Sources of concern for displacement include gentrification, being outside the seawall, and the potential for increased taxes in the future to pay for the seawall. Gentrification pressures in the Neck have been well documented, with development pushing in from Charleston in the south and Park Circle to the north [42]. Rosemont residents were particularly concerned about gentrification because of the community’s convenient location near Charleston, proximity to the marsh and waterfront views, and nearby development of the Magnolia Project that is clearing contamination and bringing upscale apartments and restaurants to the area [43].


*“I think [Rosemont] will be a very popular neighborhood in the next five to ten years. Gentrification is happening everywhere, and it’s a little bit slower where we are but I think eventually we will see more and more people pop up and I don’t think that a lot of residents are gonna get paid the worth of what their land is worth and their houses are worth.”*

*“I believe that our taxes are going to be going up here soon, and those who have fixed incomes, how do you continue to…be able to live in an area like where we’re living in…”*


Another source of concern for displacement is USACE’s expressed intent to elevate homes in Rosemont in place of inclusion within the seawall. Participants are skeptical of USACE’s claim that they can elevate every home in the community and are worried that residents of homes that cannot actually be elevated will be bought out and forced to leave.


*“A lot of people fear displacement, they feel like it’s a way of gentrification for the area…because the truth is a lot of these homes are not gonna be able to be elevated and they know that, and it’s just gonna be a way to say we can’t do this, we’re gonna have to buy you out. And especially people who have been here all their lives, where are they gonna go? Even for me I love the location, it’s just so convenient to work and places that I frequent all the time, and with real estate so high right now in Charleston you can’t find another home, everything’s so expensive.”*


***Increasing vulnerability.*** Participants expressed concerns that the vulnerability of the Charleston Neck might increase as a result of the seawall being constructed downtown, for example by causing floodwaters to inundate the Neck communities during a storm surge event, and that these communities are not resilient enough to withstand increased flooding.


*“I’m afraid with that wall, that water is gonna come here.”*


People are also concerned about potentially increased flooding in the Charleston Neck because they currently report flooding in their communities as a major problem that is not being addressed now. Participants described locations in Rosemont, Garden Hill, and Union Heights that commonly experience flooding. These areas were digitized in ArcGIS to document community concerns.

Another component related to concerns over increasing vulnerability is that flooding will increase and propagate contamination from local industries and hazardous waste sites into the communities. There are two EPA-designated Superfund sites in the Charleston Neck, Macalloy and Koppers Co., both of which are on the EPA’s National Priorities List (Figure 14) [44]. The Macalloy site formerly contained a smelting plant that has contaminated Shipyard Creek and the surrounding environment with chromium. The Koppers site was the location of a wood treatment facility that has contaminated groundwater and soil with creosote, a wood preservative chemical. There are also several active industrial sites that have experienced chemical releases in the past including Rhodia, Lanxess Corporation, Concrete Supply Co., and Chevron (Figure 14) [45]. Extreme flood events and storm surge have the potential to disturb contaminants in Superfund sites and industrial facilities, which can threaten local health and life expectancy [46,47]. If flooding is expected to increase near these communities, then pollution and contamination could compound threats to the communities’ health and safety.

A third concern related to increasing vulnerability is that the elderly and disabled would not be able to access their raised homes in Rosemont if USACE is able to elevate them. For example, 22% of the resident population in Rosemont is over the age of 65 compared to just 12% in peninsular Charleston [38]. People were concerned that climbing stairs or a ramp could be difficult for the elderly and disabled, and that if a flood event occurs that people will be afraid of being trapped in their raised homes.


*“I think we pose a really strong issue for our seniors who have bad knees, bad hips. Even though three feet doesn’t sound like a lot, that’s an extra two to three steps that someone has to climb in order to get in and out of their house…”*


***Sense of exclusion.*** Interview and field observation results showed that participants felt separated from the larger Charleston/North Charleston community. This sense of exclusion stemmed, in part, from being left outside of the proposed perimeter seawall, as people felt this position further increased their isolation and separation from peninsular Charleston.


*“...we don’t want to leave anyone behind so let’s not leave Rosemont behind, that’s where my concerns are, that’s where the residents’ concerns are…”*


People also believed that their communities were not receiving the same level of upkeep and investment as other parts of Charleston/North Charleston, with concerns going back to inadequate drainage systems and lack of access to grocery stores and other amenities such as bus shelters and sidewalks.


*“It used to be a really beautiful community…but it seems like as the years have gone by the City of Charleston…they don’t keep up with the community and our needs…I feel like we are a forgotten community.”*


This pervasive sense of exclusion may have origins in the isolation imposed by previous and current highway construction through and around the Charleston Neck communities (Figure 10 and Figure 11). 

***Mistrust of government.*** None of the interview participants could think of anything undertaken by their local governments to mitigate flooding in their communities, despite several people saying they had reached out for assistance. This created an environment of mistrust that existed well before the USACE announced the seawall plan, which may have contributed to some of the residents’ concerns about the project. Specifically in Rosemont, USACE officials did not initially recognize that the community had a flooding problem and pointed towards the community’s overall elevation being higher than downtown Charleston as evidence that the community did not experience flooding. Frustrated residents then began documenting flooding in Rosemont, compiling dozens of photographs of flooding to convince government officials that the community does experience flooding and that being left outside of the seawall could exacerbate this problem. Rosemont residents also felt that the USACE’s plan to elevate all homes in the community as part of its nonstructural flood mitigation strategy was meant to appease the community, but if they actually start elevating homes some would not be structurally sound enough to be raised, and that those residents might have to leave Rosemont.


*“If they could successfully elevate the homes then that would be great, I’m all for it…I just really don’t see it happening, I think the Corps will probably say in the end that it’s too expensive to do.”*


***Civic engagement.*** Despite the mistrust participants felt towards government officials, there was also a clear emphasis on civic engagement as a means for community advocacy. Residents seem to recognize the roles and responsibilities of local government in preserving and improving their communities. As a result, they are involved with neighborhood associations, City of Charleston advisory boards, and LAMC and CCRAB work, with the intention of improving their communities and the lives of their fellow residents. One Rosemont community leader stated:


*“The changes I hope that with each passing generation some sense of newness will come about, with different social groups that will come in and offer some sense of guidance to a better place that will take hold and start elevating, especially the youth, making quality of life better for them. With the community council, it’s about trying to be informed about the different things the city is implementing or wants to implement, and how we can get into that and be a part of it and garner some of this sense of opportunity to help us grow and help us expand…”*


For many residents, their own personal histories were drivers for their involvement in advocacy organizations and local government, with one interviewee attributing his engagement to the community work of his grandfather, uncle, and mother. The Neck communities are historic communities of color, some with ties to the settlement of enslaved African Americans in Charleston freed after the American Civil War in 1865 [24]. The Reconstruction era saw the continued development and expansion of the Neck communities until they were eventually incorporated into Charleston County and in either the cities of Charleston or North Charleston through the 1990s. The following quote from a community leader of Union Heights refers to the period between Reconstruction in the late 1800s and formal incorporation into local political boundaries through the 1900s when the Neck communities were largely self-sufficient:


*“One of my uncles worked really hard in helping the community. He pushed for the drainage to be modified and updated. Pushed for having the streets paved. Because of his activity and my grandfather, my grandmother and my mother because of their civic involvement we got to know a lot of people…A lot of people were just barely literate or totally illiterate. My grandfather was also a notary so people would come to him to get documents notarized, but they also came to him if they had an important paper, they brought it to him so he could read it for them. Seldom, people made legal decisions without consulting with him. And then my uncles, and eventually me and my daughter, fell in that role.”*


Participants expressed a desire for inclusion and more transparent conversations with their political representatives to better understand and be more involved in decisions surrounding flood mitigation. They feel that increased engagement would help address their concerns about the seawall and help build trust between local government officials and the communities. Rosemont residents have started engaging with USACE and Charleston city officials, though engagement has been limited to a few public meetings and has not yet extended to other Neck communities.


*“I would like to see…a real opportunity for us to sit down with our councilmen…and see…why they have discluded Rosemont from the report, from the seawall, and it’s kinda just have the opportunity to have an open discussion…that’s something that would make residents feel a lot safer and also understanding where their thought process is coming from.”*


Discussions about the seawall prompted conversations about additional topics such as potential benefits, concerns, questions, perceptions of resilience, and suggestions. Overall, concerns related to the proposed seawall include possible exclusion from mitigation that is intended to benefit the rest of Charleston, potential impacts to the surrounding marsh as a result of wall-associated erosion, likely accessibility issues for the disabled/elderly if homes in Rosemont are elevated, fear of being displaced from homes, and the possibility of increased taxes, many of which fall into themes related to displacement, increasing vulnerability, sense of exclusion, and mistrust of the government (Figure 15, Table 4).

Interviews and field observations provided insight as to what would be useful to include in the spatial component of this study. These included interactive elevation layers, the compilation of flood-risk locations and important places, updated elevation of the newly paved roads and sound wall in Rosemont, locations of industrial facilities and Superfund sites, and others. These data visualizations were mapped in ArcGIS Pro and included in the Story Map and were suggested and informed by participants. An overview of environmental-justice issues in the Neck communities, the progression of USACE’s proposed seawall project, and other background information were also included for context. Screenshots of the Story Map and key data layers (Appendix A) are included in the Appendix A. The Story Map format was selected to communicate spatial and interview results to everyday people using a clear language explanatory narrative. The Story Map can be accessed conveniently via a browser at https://storymaps.arcgis.com/stories/850419153fb7414391c692fc3a0a794f (last accessed on 4 September 2022) without the necessity to use GIS software.

### 3.2. Report-Back Meeting

Key study findings, the ArcGIS Story Map, and links to all materials were presented to community members and study participants at a public meeting held in Rosemont in late March 2022. The study results were received positively, with participants providing additional comments, verification, and other feedback. Attendees indicated that project outputs would be helpful to the communities for future advocacy work.

## 4. Discussion

The Earth’s coastal areas house a significant portion of the global human population, and many of the world’s largest and iconic cities are located along coasts where they face growing and existential threats associated with flooding resulting from rising sea levels, increased intensity and frequency of extreme weather events, and poor land-use decisions [1]. Climate mitigation strategies are essential, but even if adopted promptly on a massive scale, results likely will be too slow to meaningfully affect nearer-term threats to coastal cities, meaning that adaptation is an immediate necessity. Unfortunately, adaptation options to protect coastal cities are relatively limited and primarily include some forms of built or nature-based protective structures, elevation of threatened buildings and infrastructure where feasible, and retreat. In many cases, only a relatively small fraction of threatened infrastructure can be elevated cost-effectively and while retreat for small and critically endangered communities may be reasonable, for entire cities it is difficult to contemplate [3].

Like numerous other coastal cities, Charleston, SC, is considering options to reduce impacts from increasingly frequent tidal flooding and projected storm surges. A major project under consideration is partnering with the USACE to construct a massive seawall around the city. Part of the city’s deliberations involves interaction with the public about the proposed wall and what it may mean for Charleston, including both possible positive and negative effects. In its deliberations, it is imperative that the city include all of its residents, with particular attention given to those in low-wealth, predominantly minority communities that have been frequently overlooked and marginalized in the past. Not surprisingly, marginalization themes [48,49] were prevalent in the interviews and field observations recorded here. A key concern is: “How can lack of fairness in impact and inequitable distribution of costs, benefits, and risks associated with [climate change] responses be better understood and prevented?” [50].

Another central question is whether the anticipated benefits of the proposed seawall will outweigh the likely unintended negative consequences. While a detailed review of seawall effects is beyond the scope of the present study, it is well known that seawalls can be beneficial or injurious, or both, depending on local circumstances. For example, seawalls can protect people and property from flood injury, damage and displacement; preserve or perhaps increase property values; and maintain the local tax base by avoiding retreat [51,52,53,54,55,56]. In contrast, negative effects of seawalls include heightened erosion in areas adjacent to the wall ends, harm to native species, loss of biodiversity, damage to adjacent properties and their values, and adverse changes in flood dynamics [57,58,59,60,61,62]. If the proposed surge wall has potential to contribute even slightly to additional flooding in the Charleston Neck communities, it is essential to understand how that risk could change and how people’s lives, properties, and finances could be affected. Experiencing impacts to daily life as a result of these or other effects could constitute an environmental injustice. Thus, the present study was undertaken as an initial step to ensure that voices of potentially impacted environmental justice communities are heard, recorded, and communicated in their own words; demonstrate the value of locally-derived data; and provide an example of how other such proposed projects might be evaluated via an environmental justice lens.

This project leveraged multiple forms of community engagement and co-production strategies to paint a broad picture of Charleston Neck residents’ opinions and concerns related to the proposed downtown seawall and flood resilience, in general. The study’s strengths lie in the multiple avenues of community engagement and commitment to produce a record of community sentiments in a form that should be usable by LAMC, the communities themselves, and others, to make Charleston more resilient. A critical first step was to build relationships and establish trust with community members. This was accomplished through a participant-observation practice of attending meetings, participating in the EJ Strong capacity building project, and consistently communicating and connecting with community members. Representing community voice was key in this study. One example is that participants expressed a strong desire to pursue resilience as they define and perceive it, and not necessarily how experts and academics do. Thus, we used the term but did not suggest a particular definition. Another major strength of this study was being able to conduct the work alongside members of LAMC and CCRAB, which helped legitimize the project in the eyes of the communities and assisted us in connecting with participants. This study also benefited from several other efforts underway in the communities, including photovoice projects to collect evidence of flooding, advocacy work with the SC Coastal Conservation League to develop a Rosemont-specific resilience plan, and the EJ Strong disaster-risk reduction workshops, among others. These related activities provided many opportunities to engage with residents and leaders, become part of conversations surrounding environmental justice and climate change, and connect with interested and informed participants. Finally, the study connects community-derived information and opinion with geospatial data, enabling objective visualization and corroboration of participants’ concerns, thus helping equip the communities with customized data to support their advocacy efforts. This is an essential element that was generally lacking in relation to the seawall prior to this study.

An important limitation of this study is the relatively small sample size of community member interviews. While we attempted to schedule more interviews, Covid pandemic-related factors, including substantially reduced ability to gather in person, reluctance of some non-community members to be interviewed, scheduling difficulties, and time available for the work restricted our ability to do so. Nevertheless, the richness of input received from participant interviews, augmented with information from participant-observation, as well as from secondary sources such as census records and spatial data, allowed this project to proceed in its aim of knowledge co-production. Another limitation of this study is that the majority of interview participants and field observations are representative of the Rosemont community. Nevertheless, slightly more than 40% of those interviewed were from other Charleston Neck communities. Since Rosemont falls in the City of Charleston and thus the USACE’s study area, it has received more attention and communication from both and, therefore, its residents are more engaged with and informed about the seawall project. Other communities are geographically close to Rosemont and they may be affected by the wall in similar ways, but at the time of the participant-observations, did not engage as much with this project. Additionally, all of the other Neck communities fall within the jurisdiction of the City of North Charleston, which has not been involved significantly with the seawall project, to date. While the study results may be most applicable to Rosemont, all of the Neck communities were represented in spatial-data collections, some had participants at one or more of the community meetings we observed, and all should be able to develop similar tools to leverage, as needed, based on our results.

A significant finding of the study is that residents of Charleston Neck communities view themselves as parts of a social-ecological system with interconnected and complex interactions among their human, physical, and ecological components, and with unique features and challenges that differentiate them from other parts of Charleston. Since participants had such strong connections to the Neck as a treasured place to live and way of life, any threat to the Neck’s social-ecological system represents a threat to individuals’ identities, that is, how they define themselves. Thus, understanding their concerns about the potential effects of the proposed seawall is essential if the integrity of the Neck communities and their residents is to be preserved. Unfortunately, it appeared that the USACE’s environmental justice review, which is embedded within the seawall project’s EIS process, did not provide for an early and holistic view of the proposed wall, lacked adequate community engagement and locally-derived data, and was too narrow in scope to fully capture potential impacts to the Neck communities’ social-ecological system. A recommendation for the City of Charleston and USACE officials is to engage as much as possible with the Neck communities. Some important progress in this regard is already apparent in improved communication with Rosemont residents; however, to our knowledge, increased engagement has not extended to Garden Hill, Union Heights, or other Neck communities. The City of Charleston and USACE could also connect with officials in North Charleston to collaborate on flooding issues in the Neck, including those that may be influenced by the seawall, as well as others. Such engagement will require being consistent and transparent with communication and follow through with discussions and commitments in order to develop trust. Engagement should occur within the communities at locally known gathering places and at times that are convenient to residents’ schedules. There are also many opportunities for engagement to go beyond public meetings, and officials would benefit by providing additional means for resident involvement, such as phone calls, newsletters, email chains, and working groups, among others. For their part, the Charleston Neck communities are likely to persist in demanding to be heard throughout the progression of the proposed seawall project and other issues facing them. By submitting public comments, asking questions at public meetings, collaborating with LAMC, and communicating with political representatives, community members can be powerful advocates for their concerns and keep them in the forefront of city-wide deliberations. These interactions may also lead to increased collection of and interest in local-scale data that would provide more realistic assessments of potential impacts to individual communities. The Neck communities may find that the proposed seawall could serve as a platform to increase engagement on other issues affecting them as well.

In addition to the above-mentioned activities, perhaps the most significant opportunity for future research that was identified is that offered by the establishment of strong, long-term relationships between community-based organizations (CBOs) that represent and reflect the voices of disadvantaged, underserved, and marginalized communities and academic institutions that are committed to community-based participatory research. CBOs can include independent community organizations, faith-based groups, health centers and community health workers, and many others. Working together, such partners can identify and address community needs, hazards, vulnerabilities, and capacities. As demonstrated here and by Palinkas et al. [63] and Springgate et al. [64], CBOs such as LAMC not only may have a wealth of resources for partnering with researchers, but more importantly can connect researchers with community leaders and members, provide logistical and other support for community-based research and knowledge co-production, introduce academic researchers to locals who have significant traditional ecological and cultural information (i.e., “PhDs of the sidewalk”) [64], provide opportunities for student internships and research, and help legitimize the presence and activities of academics in their communities. Among many other things, academic researchers can connect local knowledge with new geospatial data and tools to assist communities in visualizing problems and issues and envisioning better futures. Investments by academic institutions in long-term relationships with CBOs will provide a strong foundation for community-based academic research that can make enormous differences in development, acceptance, and implementation of useful and usable science to enhance community resilience in the face of global change. Robust, long-term alliances between communities and academic institutions can create compelling forces for positive changes that last far beyond those accomplished via the usual project-by-project frameworks by which research is typically conducted.

## 5. Conclusions

As climate change continues to worsen flooding in coastal communities, additional inequalities and vulnerabilities of lower income and communities of color will be exposed. Therefore, it is important to leverage opinions and concerns of people in such communities to increase resilience in meaningful, equitable ways. The City of Charleston and the USACE have the opportunity to use the peninsula study and proposed seawall project as a bridge to enhance engagement with the Charleston Neck communities. While engagement has been pursued, to some extent, in Rosemont, there is still a need to engage with all of the Neck communities, including those that fall within the jurisdiction of the City of North Charleston. Residents in the Neck communities already feel isolated and excluded from the greater Charleston area, and these feelings could be addressed positively through intentional, meaningful engagement and follow-through. As one of the first East Coast cities pursuing major structural mitigation for flooding, Charleston will become a model for other cities as waterfront mitigation becomes necessary for more and more areas. Meaningful, representative and inclusive engagement are important to ensure that both benefits and costs of mitigation measures will be equitably distributed across the area’s breadth of socio-economic, racial, ethnic, and other groups. Bringing more people to the table to hear their opinions and concerns and creating vibrant, long-term partnerships between academic institutions and CBOs that include robust links to governmental organizations should be among the first steps towards more inclusive, equitable resilience-building.

## Figures and Tables

**Figure 1 ijerph-19-11192-f001:**
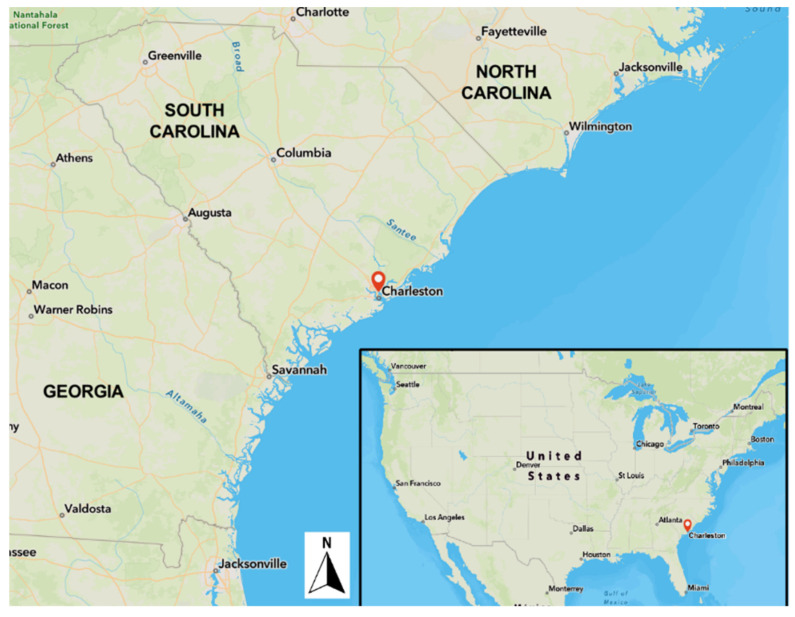
Map showing location of Charleston, SC, on the US South Atlantic coast (prepared by J. Taylor using ArcGIS).

**Figure 2 ijerph-19-11192-f002:**
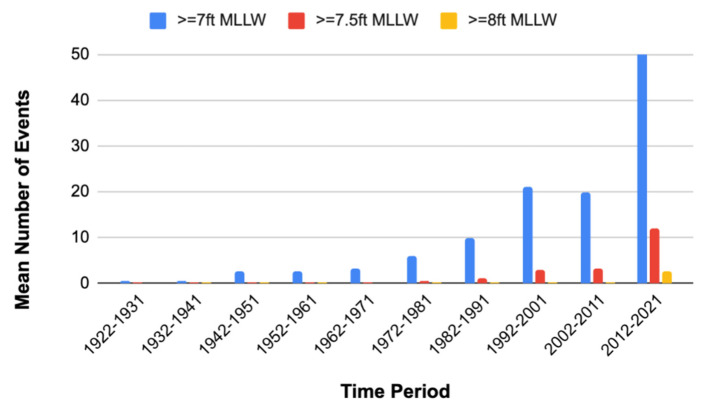
Mean number of very high tide events in Charleston Harbor, SC, by 10-year intervals from 1922–2021 (based on data from [7]).

**Figure 3 ijerph-19-11192-f003:**
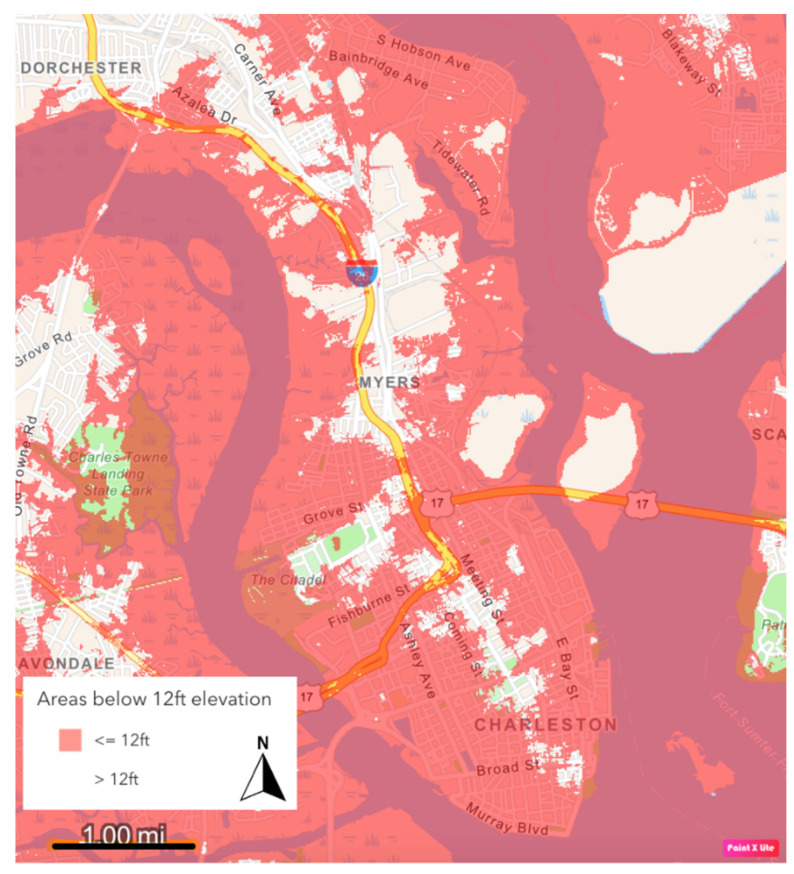
Parts of the Charleston peninsula that are <12 ft (3.7 m) elevation (in red). Digital Elevation Model (DEM) developed by the Lowcountry Hazards Center at the College of Charleston.

**Figure 4 ijerph-19-11192-f004:**
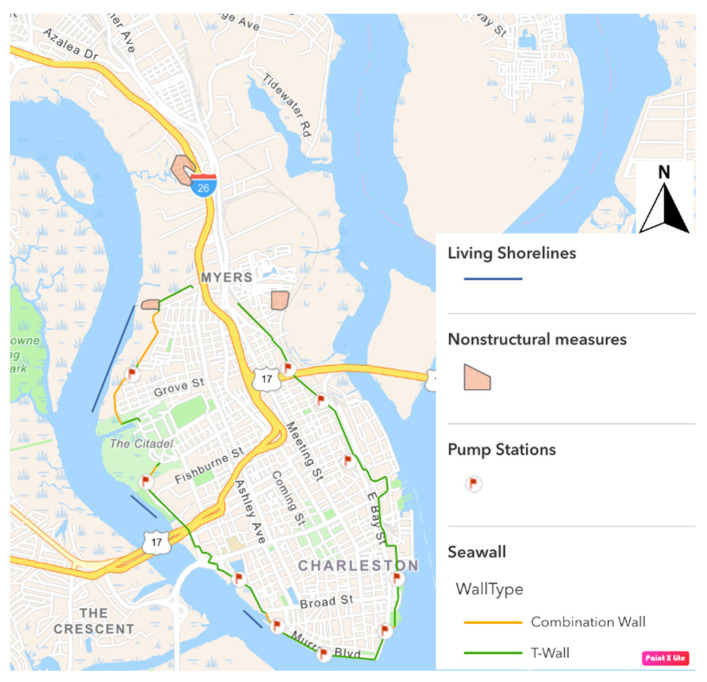
Map of proposed surge wall location on the Charleston, SC peninsula, digitized from the USACE NEPA (National Environmental Policy Act) Scoping Meeting Presentation [15]. Green lines represent T-Wall locations, orange lines represent combination wall locations, pink polygons represent nonstructural measures such as floodproofing or elevating homes, red flags represent locations of pumping stations, and blue lines represent locations of living shorelines.

**Figure 5 ijerph-19-11192-f005:**
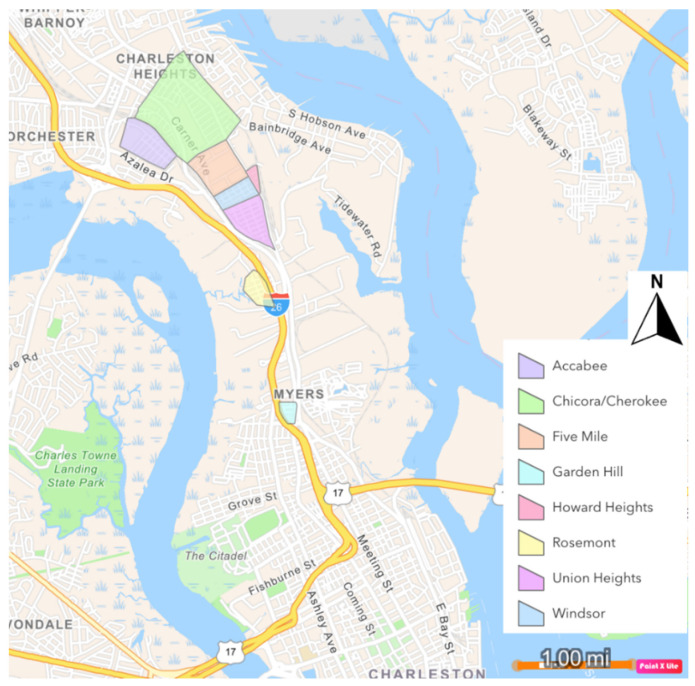
Approximate locations of LAMC communities in the Charleston Neck (digitized from descriptions in LCRT’s NEPA report [17].

**Figure 6 ijerph-19-11192-f006:**
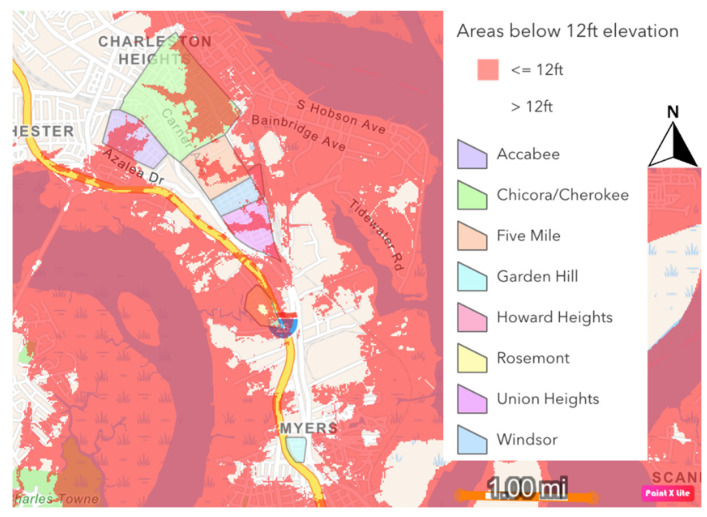
Parts of the Charleston Neck under 12 ft (3.7 m) elevation (in red), in relation to Charleston Neck communities. Digital Elevation Model (DEM) developed by the Lowcountry Hazards Center at the College of Charleston.

**Figure 7 ijerph-19-11192-f007:**
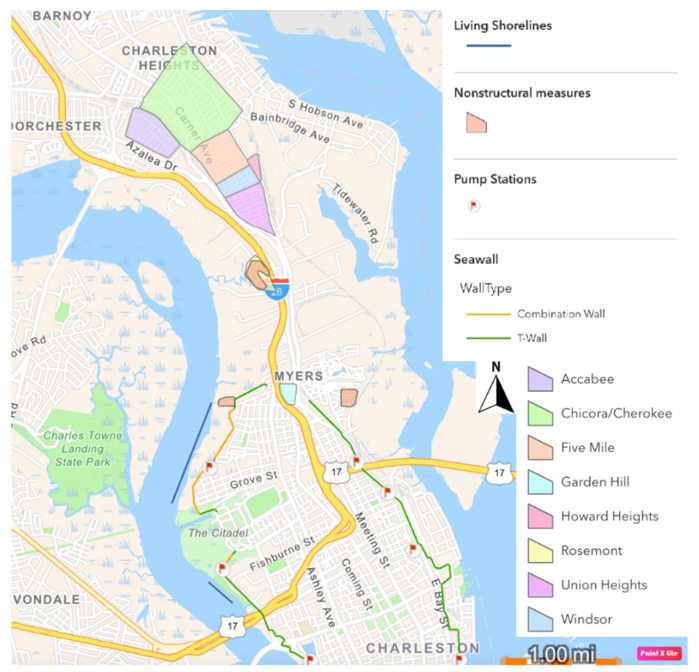
Location of proposed seawall in relation to Charleston Neck communities.

**Figure 8 ijerph-19-11192-f008:**
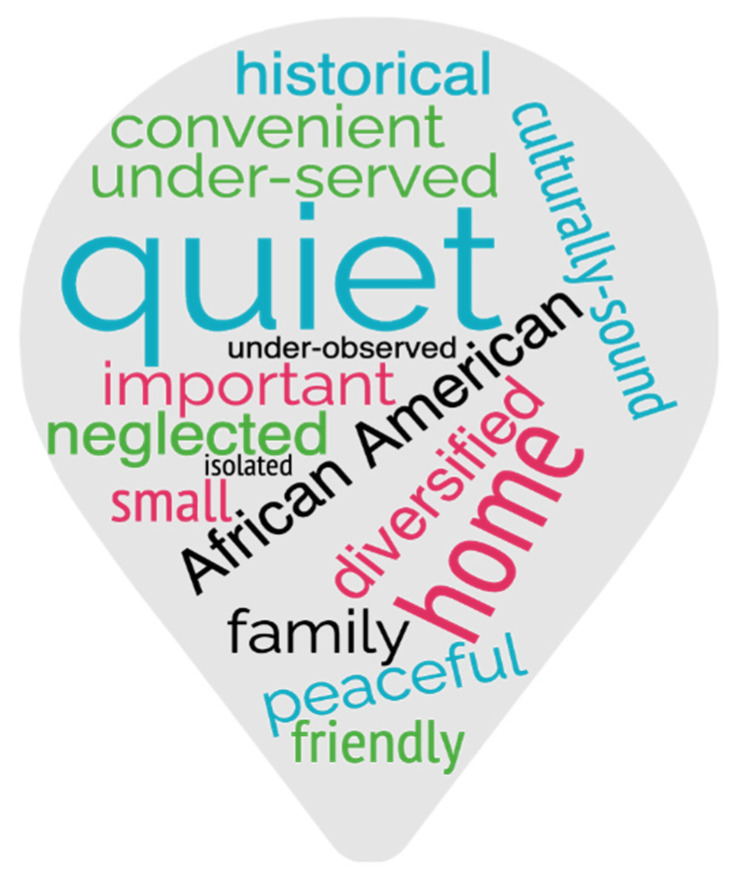
Word cloud of terms used by participants to describe their communities [39]. Interview sample size was low, so frequencies are not assumed to be generalizable to all Neck residents.

**Figure 9 ijerph-19-11192-f009:**
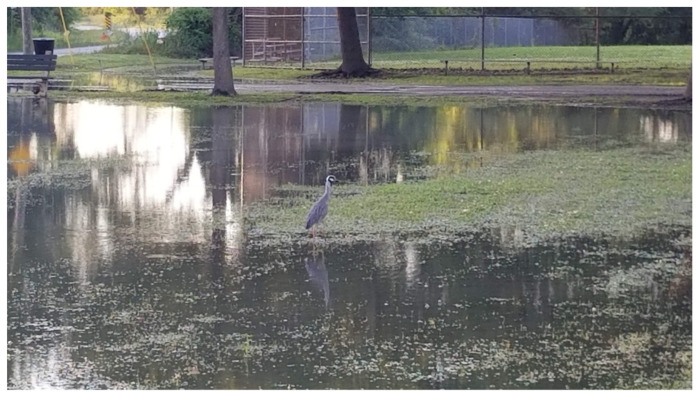
Flooding in Rosemont Park on 24 April 2020, photograph taken by resident and provided to the research team.

**Figure 10 ijerph-19-11192-f010:**
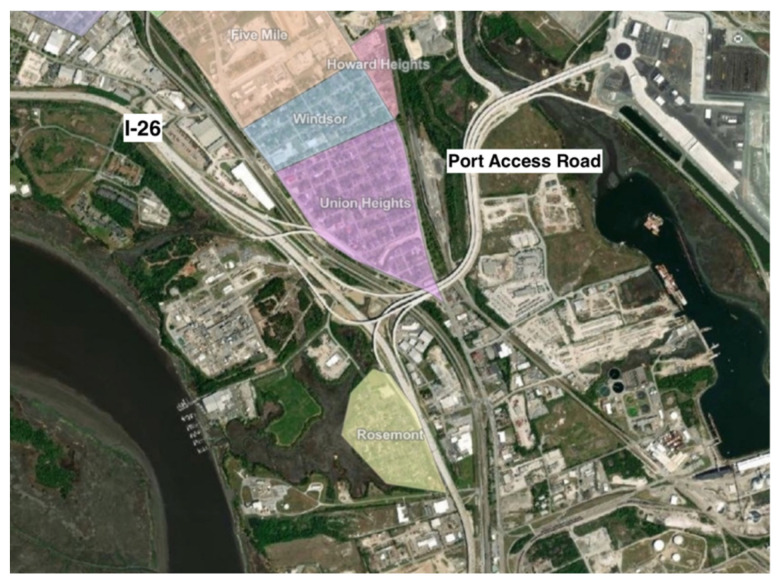
Aerial imagery showing the network of highway infrastructure surrounding and separating the Rosemont and Union Heights communities.

**Figure 11 ijerph-19-11192-f011:**
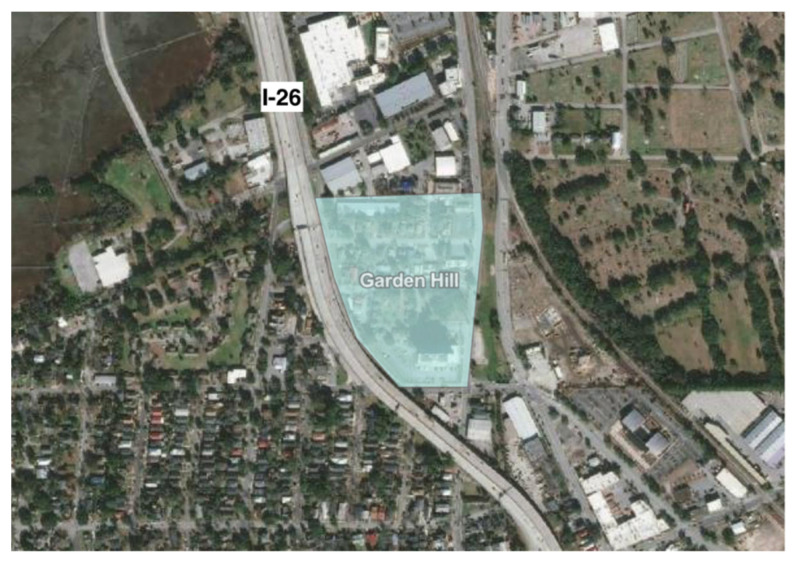
Aerial imagery showing highway infrastructure surrounding the Garden Hill community.

**Figure 12 ijerph-19-11192-f012:**
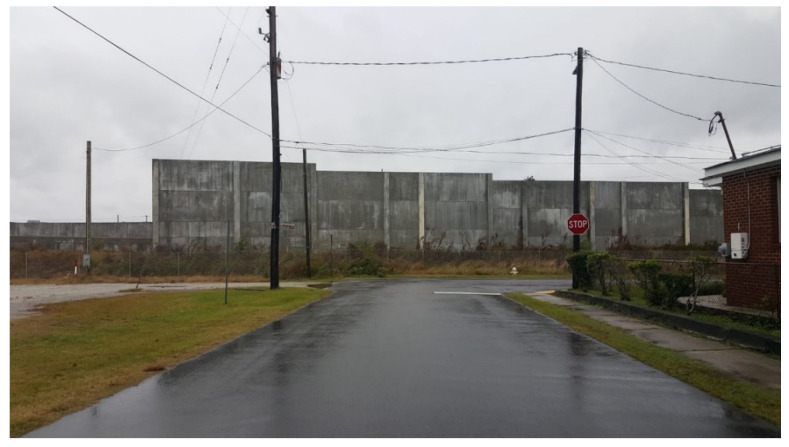
A portion of SCDOT’s sound wall in Rosemont, photograph taken by Judy Taylor on Peace Street.

**Figure 13 ijerph-19-11192-f013:**
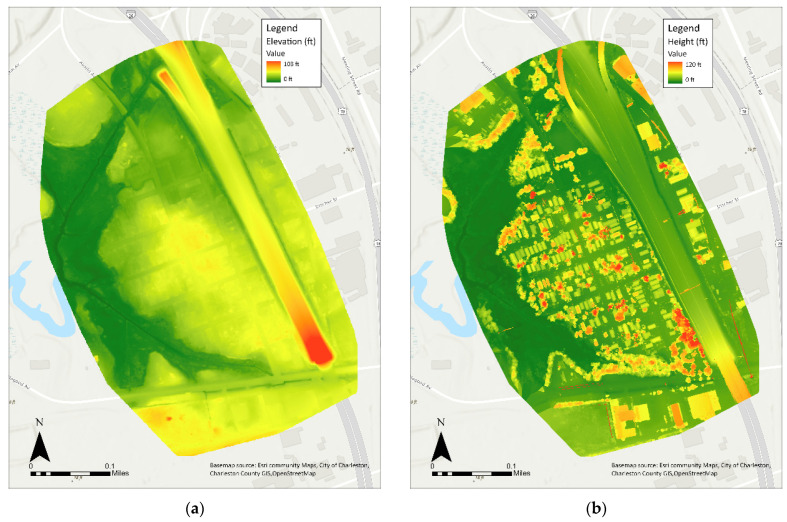
(**a**) Lidar-based Digital Elevation Model (DEM) depicting elevation in Rosemont elevation. Elevation data are in feet above sea level [NAVD88]. (**b**) Drone-based Digital Surface Model (DSM) depicting elevation and building heights in Rosemont. Elevation data are in feet above sea level [WGS84]. (Lidar and Drone data Developed by the College of Charleston, Lowcountry Hazards Center). Black line in (**b**) alongside I-26 is the new sound wall shown in Figure 12. The I-26 highway is shown entering the bottom right corner of each figure and proceeding diagonally across the figure.

**Figure 14 ijerph-19-11192-f014:**
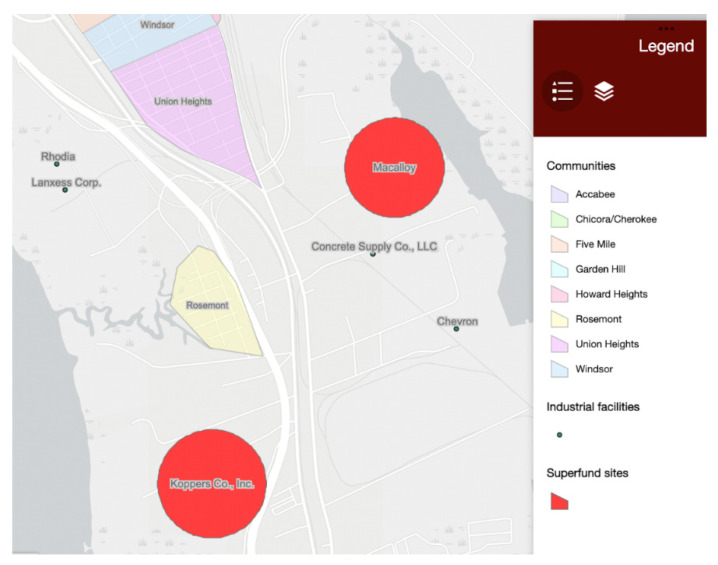
Locations of Superfund sites (large red circles) and industrial facilities (smaller black dots) that have experienced releases in the Charleston Neck [44,45].

**Figure 15 ijerph-19-11192-f015:**
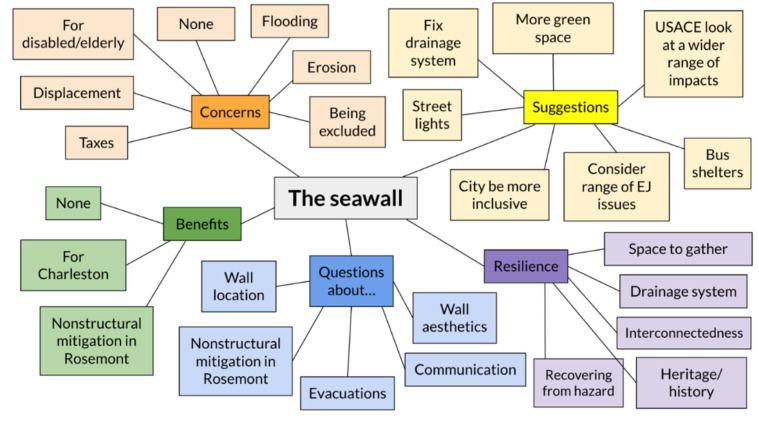
Thematic map depicting the range of responses prompted by discussion about the seawall and arranged according to major topics (Concerns, Benefits, Questions, Resilience, and Suggestions).

**Table 1 ijerph-19-11192-t001:** List of prepared questions for interviewees.

Question	Interviewee
What year were you born?	Community resident/leader
Tell me about your community; can you describe your community in three words?	Community resident/leader
How long have you lived here?	Community resident/leader
Why is this place important to you?	Community resident/leader
In what ways has your community changed in the time you’ve lived here?	Community resident/leader
Take me through the average week in your life; where do you go on the Charleston Neck? What do you leave your community for?	Community resident/leader
Do you fish or gather plants, or know of community members who do? What animal or plant species do you or others target?	Community resident/leader
Tell me about the last time your community experienced a flood (if it ever has). Can you describe this experience? Can you describe where this flood occurred?	Community resident/leader
Can you tell me about the most severe flood your community has experienced?	Community resident/leader
How are people’s daily lives affected by floods?	Community resident/leader
Have you noticed a change in the frequency or severity of floods since you’ve lived here?	Community resident/leader
What has been done by either the community, the city, or the County to reduce flooding in your area?	Community resident/leader
Can you think of ways your community would benefit from construction of the wall?	Community resident/leader
Do you have any concerns about potential negative impacts from the wall on your community?	Community resident/leader
What would you like to know about the proposed seawall and its potential effects on your community?	Community resident/leader
What types of information, maps, reports, or presentations do you think would be helpful for us to include in the project?	Community resident/leader; City official
How would you like to receive the results?	Community resident/leader; City official
Do you have any questions or general comments for me?	Community resident/leader;City official
How would you describe a “resilient” Charleston?	Community resident/leader; City official
Which parts of the City would you say are the most vulnerable to flooding?	City official
Outside of the wall, what steps is the City planning to take to reduce the effects of flooding in the upper part of the City, including Rosemont?	City official
Are there any plans to work with North Charleston on understanding impacts of the seawall on the other communities in the Charleston Neck, like Union Heights, Windsor, etc.?	City official
If City Council doesn’t approve the wall, what alternatives has the City considered to manage flood risk besides constructing the wall with the Army Corps?	City official
How do you see the upper part of the City benefiting from the seawall project?	City official
Has the City considered how the seawall may affect adjacent areas in the upper part of the peninsula, beyond the Army Corps’ surge modeling? What, if anything, do you think those effects would be?	City official

**Table 2 ijerph-19-11192-t002:** Community participant affiliation and demographic information.

Demographic Characteristic	Coding	N (%)
Age	30–50 years	3 (37.5)
	50+ years	4 (50.0)
Affiliation	Rosemont	4 (50.0)
	Union Heights	2 (25.0)
	Garden Hill	1 (12.5)
	City of Charleston	1 (12.5)

**Table 3 ijerph-19-11192-t003:** Codes and examples used to categorize the content of interview responses and data gathered from community meetings and other sources in NVivo. We use italics throughout the paper to clearly identify quotes from study participants.

Content-Based Code	Sub-Code	Illustrative Quote
1–3 word description of community	Positive descriptors	*“important”; “home”*
	Negative descriptors	*“isolated”; “underserved”*
Changes in community	Attitudes/activities of residents	*“…people started moving out and people came in renting…some people if they’re renting it’s not theirs so they don’t care about it.”*
	Demographics	*“Before it was an all African American community and now you see other ethnicities moving in the neighborhood.”*
	Upkeep	*“The City of Charleston…they don’t keep up with the community and our needs like we need them to.”*
	Amenities	*“But there were a lot of things this community had…like a little grocery store, we had cleaners…businesses and stuff in this community, you know all that changed.”*
	Infrastructure	*“The only major change I see is that wall that they put for the exit, supposed to be a soundwall.”*
Why is community important	History	*“...there’s not many neighborhoods left like Rosemont…it still has the historical presence, and a lot of the people are descendants of those who started our neighborhood…”*
	Residents	*“We have a lot of elderly people that are very very friendly, and they also look out for one another here.”*
	Grew up there	*“It’s where I grew up, it’s home.”*
	Quiet	*“It’s nice and quiet…”*
Fishing	Importance of fishing	*“I fish a lot, that’s something I’m really trying to teach my daughter is how to live off the land”*
	Downtown (Charleston)	*“My brother does, he goes downtown fishing”*
	At Northbridge Park	*“...we went over north bridge here and right along there we went there to go fishing.*
	Fish species	*“Spot tailed bass, what we call red drum, and whiting and tell you what, they had some humongous catfish in there.”*
	Types of fishing	*“crabbing, shrimping, fishing”*
Gardening	Concerns about contamination	*“I want to grow a garden…but with that chemical plant so nearby I’ve always had concerns about growing food and actually eating it here in this neighborhood”*
	Gardening in the past	*“[Name redacted] had implemented a garden a few years back…for a couple of seasons that went really really well, they were feeding off it, but then it kind of just died down.”*
Transportation	Bus	*“We have people who use the bus system…”*
	Walking	*“I walk around a lot, walk to the park over there, to the beginning of Rosemont then back to the end of Rosemont”*
	Driving	*“I do leave; I leave here every day for work and my son attends school on James Island so we have to take him to school and pick him up”*
Important places	Community center	*“...our community center is very important…”*
	Grocery store	*“Food Lion”*
	Convenience stores	*“…we have a couple convenience shops within the neighborhood, one is GoGos…the next one would be the gas station on the other side of Meeting Street”*
	Churches	*“churches in Rosemont”*
	Downtown Charleston	*“I think everybody does most of their living downtown.”*
	Marsh	*“the marsh right there in Rosemont”*
	Rosemont Park	*“People do like to come and just kinda sit in the park.”*
Personal stories	-	*“One of my uncles worked really hard in helping the community”*
Miscellaneous concerns	Gentrification	*“…that leaves then Rosemont barren, which opens up the door for gentrification…”*
	Characterization that community does not flood	*“The community has expressed concerns with flooding within their neighborhoods, and they are worried about the characterization their communities do not flood.”*
	Drainage system	*“We need them to fix the drainage system around here”*
	Traffic	*“People come speeding through here all the time all day long”*
	Tree Maintenance	*“We got trees that need to be cut”*
	Racial discrimination	*“It’s one of those communities that’s culturally underserved”*
	Lack of transportation	*“If they don’t have a car, they’re stuck”*
	Lack of water access	*“To not have [access] I think deprives them of the amenities of living there along the water and the marsh.”*
	Contamination from local industries	*“We are in the factory area, so that could be very dangerous with runoff from the factory and that going back inside of the marsh and how that could affect our people.”*
	Economic hardship	*“…the one thing that’s held our people back is the lack of generational wealth.”*
	SCDOT’s sound wall in Rosemont	*“I don’t feel like it’s helped us, when you’re outside you can still hear all of the traffic on the interstate. I hate how it looks aesthetically for the neighborhood, it looks awful.”*
Flooding	Experiences with floods	*“...my dad used to have to pick me up to get to the door to put me in the car, we always had boots…”*
	Road paving in Rosemont	*“’Now because of the paving in the street… there’s this little like gully that’s filled and the water spills over the sidewalk over to my property…’”*
	Highway construction	*“They were doing construction on the overpass…that might have caused streets that were flooding that didn’t get flood”*
	High tide flooding	*“If they get a high tide [Rosemont] can have flooding”*
	Sources of flooding	*“That’s one source for Rosemont, the marshes. The other source is the higher streets and the third is the sound wall.”*
	Wildlife drawn to floods	*“I don’t love the sound of hearing frogs and crickets or whatever other creatures are out there cause of the water.”*
	Mosquitoes	*“there’s mosquitoes breeding right in my yard”*
Changes in flooding	Increased flooding	*“...it has become a major concern, all the years I’ve been back here it’s never flooded like this back here.”*
	Drainage	*“I think it goes back to if the drains are cleaned regularly on a regular basis”*
	SCDOT sound wall and road construction	*“Once they added that wall there and rebuilt that road a lot of that stormwater runoff…it’s come down into the neighborhood, because before there was trees and stuff there, and you didn’t get all that runoff.”*
Resilience definitions	Maintaining heritage	*“...we can still keep our memories and our heritage, you know, our home…”*
	Gathering space	*“a neighborhood where the people have a place to congregate like a park or something with activities”*
	Communities working together after a disaster	*“When in ‘89 when Hurricane Hugo hit here, this city was tore up, but we came together as a city, as a community, and all the communities came together and pitched in and removed all that debris”*
	Green space, drainage	*“I would like to see more green spaces, drainage improvement back here”*
	Cooperation	*“It’s gonna take…people settling differences…that’s how we can build a better, resilient Charleston in my eyes, and we gotta get our church leaders and everybody involved”*
Benefits of the seawall	Better overall protection	*“…from a project as a whole, that would give us the opportunity to protect ourselves a lot better…”*
	Preserving Charleston	*“We need to preserve this history, we need to preserve this culture, we need to look at keeping the city functional”*
	Nonstructural mitigation	Nonstructural mitigation
	No benefit	*“I don’t see any benefit to it at all, actually I feel it’s gonna be a disaster for the neighborhood”*
Concerns about the seawall	Increased taxes	*“For those that are living on fixed incomes they might not be able to make those new tax regulations…”*
	Contamination from local industry	*“Residents have expressed concerns with [nonstructural mitigation] due to the ineffectiveness to prevent and/or reduce damages attributed to storm surge impacts such as floodwater carrying pollution”*
	Increased flooding	*“I’m afraid with that wall, that water is gonna come here.”*
	Being left behind	*“We don’t want to leave anyone behind so let’s not leave Rosemont behind, that’s where my concerns are”*
	City/USACE not following through on discussions	*“The idea that we’re gonna do this later, like you’re kicking the can down the road man, you’re throwing up these distractions to say you’re gonna appease me with this nonsense”*
	USACE unable to elevate some Rosemont homes	*“Many feel their homes would not meet the elevation or flood proofing requirements, and they will be forced to give up homes they have lived in all of their lives.”*
	Elevated homes in Rosemont	*“I think we pose a really strong issue for our seniors…even though 3 feet doesn’t sound like a lot, that’s an extra 2 to 3 steps that someone has to climb”*
Questions about the seawall	Why Neck is not included	*“...you’re trying to preserve a certain area, is that one area more important than all the others?”*
	Design	*“I’m really curious as what is this wall going to look like, and how are we supposed to get out of this wall”*
	USACE’s communication	*“I would like to know what they have done to inform all the people, all the neighborhoods”*
	Impact on park system	*“where we are with our park system throughout the City of Charleston”*
Suggestions for community improvements	More green areas	*I would love to see more green areas somehow incorporated in the neighborhood.”*
	Drainage	*“The community feels strongly there needs to be an improvement to the stormwater system.”*
	Streetlights, bus shelter	*“I think we need more streetlights, we definitely need like maybe like a bus shelter”*
	Space for senior citizens	*“if they could get maybe a senior citizen place, or if they could upgrade that building so senior citizens can go in there and do activities during the day”*
Suggestions for this study	Discuss advocacy	*“…express our sense of advocacy for betterment of the community, no more no less, our just due.”*
	Pictures of flooding	*“photos of the flooding”*
	Elevation and aerial imagery	*“elevations,” “GPS and satellite images”*

**Table 4 ijerph-19-11192-t004:** Illustrative quotes for responses prompted by discussion about the proposed seawall. We use italics throughout the paper to clearly identify quotes from study participants.

Topic	Sub-Topic	Illustrative Words/Quotes
Concerns	Increased flooding	*“If you’re trying to keep the water out of this area then where is the water going to go?”*
	Difficulties for the elderly and disabled	*“I think we pose a really strong issue for our seniors who have bad knees bad hips…”*
	Displacement	*“someone said they were worried about eminent domain for those houses that physically can’t be elevated.”*
	Increased taxes	*“When they raise taxes that can cause people to lose their homes.”*
	Isolation/being excluded	*Being outside a wall will act as a barrier and will isolate communities.*
	None	*“No”*
Benefits	For Charleston as a whole	*“preserving this is commendable”*
	For the Charleston Neck specifically	*“They’re getting the same level of surge risk mitigation as the rest of the city.”*
	Nonstructural mitigation	*“I think that it would be a good thing for those who don’t have disabilities.”*
	None	*“I really just don’t think it’s gonna benefit us at all.”*
Questions	Wall location	*“How are we supposed to get out of this wall, if it only has 4 exits”*
	Wall aesthetics	*“I would like to know what the material is made up of.”*
	Nonstructural mitigation	*“How will homes be selected for elevating?”*
	Evacuations	*“If we have to evacuate during a flood, are they going to come and get the elderly”*
	Communication	*“I would like to know what they have done to inform all the people.”*
Resilience	Space to gather	*“A neighborhood where the people have a place to congregate…”*
	Interconnectedness	*“I think that if we can work together… that’s how we can build a better, resilient Charleston”*
	Heritage/history	*“We can still keep our memories and our heritage you know, our home”*
	Recovering from a hazard	*“When Hugo came and the idea what we’re gonna do, how we’re gonna do this, and minds came together…”*
Suggestions	Fix drainage system	*“...we need them to fix the drainage system.”*
	More green space	*“I would like to see more green spaces”*
	USACE to consider a broader range of impacts	*“Have some vision to say, how is it going to impact the farthest reaches of the Ashley River all the way to [Bacons] Bridge Road in Summerville and beyond.”*
	City of Charleston to be more inclusive	*“Include us now in the dynamics of what has to be done …”*
	Considering a range of EJ issues	*“Residents have expressed concerns with [nonstructural mitigation] due to the ineffectiveness to prevent…floodwater carrying pollution”*
	More streetlights	*“I think we need more streetlights.”*
	Bus shelters	*“We definitely need maybe like a bus shelter, ‘cause we got the sign, but we ain’t got no bus shelter.”*

## Data Availability

Data and other supporting information are deposited with the Lowcountry Alliance for Model Communities and can be accessed via lamcinfo18@gmail.com or by contacting the first author at taylorj4@g.cofc.edu.

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
