# Peer review of "Participatory and Spatial Analyses of Environmental Justice Communities’ Concerns about a Proposed Storm Surge and Flood Protection Seawall"

_ijerph, 2022, doi:10.3390/ijerph191811192_

Round 1

Reviewer 1 Report

This paper provides results of an EJ community that expressed concerns regarding exclusion from the coastal structural sea wall to protect against flooding. The paper is written in a concise manner and is to the point, which makes it an impressive piece of community-engagement study. Information collected during the interviews is well represented and a detailed discussion has been provided. Community’s concerns with the exclusion from flood protections measures by the USACE is valid and must be re-evaluated. It should be noted that the sample size of the participants is small and only focuses on one of the eight communities, adding further participants would have helped. Overall, considering the interesting insights provided by this study on EJ, and to serve as an example for other US citywide coastal protection projects, I recommend minor revisions to the manuscript before acceptance.

Please see comments in the attached PDF.

Author Response

We thank the reviewer for his/her comments; they helped us improve the manuscript. Please see responses below and refer to the track-change revision of the manuscript.

 Reviewer 1:

  1. This paper provides results of an EJ community that expressed concerns regarding exclusion from the coastal structural sea wall to protect against flooding. The paper is written in a concise manner and is to the point, which makes it an impressive piece of community-engagement study. Information collected during the interviews is well represented and a detailed discussion has been provided. Community’s concerns with the exclusion from flood protections measures by the USACE is valid and must be re-evaluated. It should be noted that the sample size of the participants is small and only focuses on one of the eight communities, adding further participants would have helped. Overall, considering the interesting insights provided by this study on EJ, and to serve as an example for other US citywide coastal protection projects, I recommend minor revisions to the manuscript before acceptance.

Authors’ Response:  Thanks very much. We agree with the reviewer’s assessment and also that the primary weakness of the study is the small number of interviews we were able to complete due to difficulties associated with the Covid pandemic (see lines 273-275 and 738-740). As noted, we would have preferred to have more. While the reviewer is correct that the majority of our results are related to one community (Rosemont), 3 of the 7 (43%) of the community member interviews were from other communities plus there was one additional interview with a city official (see Table 2).  We have modified the limitations paragraph in the Discussion section beginning on line 736 in the track change revision to more clearly describe study limitations.  Also, as noted in lines 292-294, 740-743, we augmented the interview results considerably with observations made at community meetings and other means. The relevance of our observations to the other communities is further discussed in lines 753-756 in the track change revision.

Reviewer 2 Report

The information in the Abstract of the paper is adequate and relevant, however authors should review the formatting of the Abstract as this is not the format of this journal. You should eliminate the topics: "Background", "Methods", "Results" and "Conclusions", and write the Abstract in a continuous form. The authors have followed the formatting of other journals, which is not correct. The authors should review this situation and correct it.

How was the questionnaire that was used in the interviews designed? How was this questionnaire initially validated? The authors state in the topic "2.2. Research Approach" (line 226-227) that the study was validated by the College of Charleston's Institutional Review Board (IRB), did this include, the initial validation of the questionnaire? This information should be clarified and- added to the article, to improve the scientific rigors of the article.

The authors' detailed explanation (line 244-251) of how the people who will participate in these interviews were approached is very well done, and leaves the scientific community in no doubt as to the validity of how they were conducted.

The author indicates which sample was selected for this research, but does not identify which criteria he took into account for the selection of this sample. It would be very important for this information to be included in the article.

The author has not identified in the article, what is the Research Question, which led him to the development of this research. What does this article want to demonstrate? What was the gap that the author identified in his Literature Review (in other scientific articles), that this research intends to fill? This question is very important to be written in the article, to demonstrate the relevance and importance of this future publication to the scientific community.

How did the author carry out the analysis of the consistency of the data obtained? What is the reliability of the data obtained from the survey? This information is very relevant, and should be included in the article by the author.

The statistical analysis of the data is very weak and should be strongly improved to enhance the conclusions to be obtained. Why were statistical hypothesis tests not done? Has any significance level analysis of the data obtained been done? Collecting the answers and giving your result only, is not what is expected just in reading a scientific paper. This article is interesting, however the author should review the answers and strongly strengthen his statistical analysis.

It would be very important that the authors at the end of the article indicate what future work can be developed from this research. It would be very good if this research did not stop here, given its potential, and that the scientific community could give it continuity, and give it the importance and relevance that this research may have in the future.

The authors should add the limitations that this research had. What did the authors want to have done during the course of this research, and for whatever reason, it was not possible? What was or what were the difficulty/limitations that the authors felt and experienced? This information would be very useful to share, to help future research, and also help interpret some of the results obtained.

Author Response

We thank the reviewer for his/her comments; they helped us improve the manuscript. Please see responses below and refer to the track-change revision of the manuscript.

Reviewer 2:

  1. Comments made directly in manuscript: Table 2:  affiliation lists 8 people in total while age rows have 7 only.

Fig. 13: Make sure the max value of color scale is same in both figures; A reference to vertical datum is also missing to properly interpret the data. Authors may add elevation above Mean sea level or NAVD88; and I-26 labeling is hiding DEM elevation, please remove or increase transparency.

Authors’ Response: The text has been modified to state that the city official was excluded from Table 2 to preserve anonymity and because different interview questions were used (see Table 1) (see lines 270-272 in the track change manuscript).  In addition, Table 2 has been retitled as “Community participant affiliation and demographic information” for clarity.

For Fig. 13, we have updated the figure, confirmed that the color scales are the same, added the vertical datum information to the figure heading, and removed the I-26 labels and added information about the highway location to the figure heading.

  1. The information in the Abstract of the paper is adequate and relevant, however authors should review the formatting of the Abstract as this is not the format of this journal. You should eliminate the topics: "Background", "Methods", "Results" and "Conclusions", and write the Abstract in a continuous form. The authors have followed the formatting of other journals, which is not correct. The authors should review this situation and correct it.

Authors’ Response: The Abstract has been modified as suggested. The last sentence in the abstract has been revised and now reads as follows: “Bringing more people to the table and creating vibrant, long-term partnerships between academic institutions and community-based organizations that include robust links to governmental organizations should be among the first steps in building inclusive, equitable, and climate resilient cities.”  See lines 38-40 in the track-change version of the manuscript.

  1. How was the questionnaire that was used in the interviews designed? How was this questionnaire initially validated? The authors state in the topic "2.2. Research Approach"(line 226-227) that the study was validated by the College of Charleston's Institutional Review Board (IRB), did this include, the initial validation of the questionnaire? This information should be clarified and- added to the article, to improve the scientific rigors of the article.  

Authors’ Response: This project did not use a survey methodology, but a community-based qualitative methodology that was triangulated between interviews and participant-observation. The interview question design was based on consultations with community leaders and through the direct participation of the LAMC organization, who ensured the relevance and validity of the qualitative data to collect. The text has been amended to reflect this (see lines 251-256 in track change revision).

  1. The authors' detailed explanation (line 244-251) of how the people who will participate in these interviews were approached is very well done, and leaves the scientific community in no doubt as to the validity of how they were conducted.

Authors’ Response: Thanks for your kind remarks and validation of our approach.

  1. The author indicates which sample was selected for this research, but does not identify which criteria he took into account for the selection of this sample. It would be very important for this information to be included in the article.

Authors’ Response: “Community members willing to contribute to this study were sampled via a snowball approach and identified through LAMC, word-of-mouth recommendations from community residents, and flyers distributed during community meetings and canvassing. Interviewees were chosen on the basis of their community affiliation, interest, and willingness to be interviewed. No attempt was made to select a “representative” sample of the populations as that was beyond the scope of the project and in accordance to community-based methodologies.” The text has been modified to clarify (see lines 257-263 in the track change manuscript).

  1. The author has not identified in the article, what is the Research Question, which led him to the development of this research. What does this article want to demonstrate? What was the gap that the author identified in his Literature Review (in other scientific articles), that this research intends to fill? This question is very important to be written in the article, to demonstrate the relevance and importance of this future publication to the scientific community.

Authors’ Response: The present study was undertaken to address the following question:  What, if any, concerns may environmental justice communities that are not included within the proposed sea wall have about how the wall and flooding might affect their communities?  We addressed this question by developing …”  (see lines 142-146 in the track change manuscript).

  1. How did the author carry out the analysis of the consistency of the data obtained? What is the reliability of the data obtained from the survey? This information is very relevant, and should be included in the article by the author.

Authors’ Response: The project did not use a survey methodology, but to ensure the reliability of the data, interview data were triangulated with the participant-observation data, and vice-versa, as well reviewed for reliability by the direct participation of the LAMC organization—all in accordance with typical qualitative and community-based methodologies.  The text has been revised to include this information (see lines 291-295 in the track change manuscript)

  1. The statistical analysis of the data is very weak and should be strongly improved to enhance the conclusions to be obtained. Why were statistical hypothesis tests not done? Has any significance level analysis of the data obtained been done? Collecting the answers and giving your result only, is not what is expected just in reading a scientific paper. This article is interesting, however, the author should review the answers and strongly strengthen his statistical analysis.

Authors’ Response: “Data collected in this study were from semi-structured interviews, not surveys, and participant observation during meetings, workshops, and other events in the communities. These are qualitative data that were not intended for standard statistical analysis for generalizability, because the researchers were searching for all possible community ideas regarding the cause and effects of flooding and plans for mitigation—an inductive rather than deductive research design. In addition, the interview sample size was insufficient to discern potential relationships. Our intent from the outset was to identify important themes via qualitative coding of interview responses and then to link them with spatial data using GIS, so statistical relevance of observed phenomena to map was not necessary. We employed a commonly used software, NVivo [33], to code qualitative responses to interview questions and notes taken during community meetings, following the approach of Cope [34]. (see lines 299-310 in the track change manuscript).

Additional explanation not included in the manuscript text: This inductive project aimed to generate theory from marginalized communities that claim not to be represented in the existing scientific literature. This community mistrust is why the research team chose to engage a knowledge co-production process and inductive research design. As such, this community-based methodology requires that researchers be open to exploring topics at the community direction, so the placement of hypotheses must be different than a survey research design. In a knowledge coproduction project, the intent is for the researchers to work with community members to generate hypotheses in the process of their work together, not at the outset of the study; the results section reviews what all these hypotheses are, and in one case the research team was able to further investigate a community observation via drone imagery. This kind of research process of community engagement is the main argument of the paper of how science working with the public ought to proceed.

  1. It would be very important that the authors at the end of the article indicate what future work can be developed from this research. It would be very good if this research did not stop here, given its potential, and that the scientific community could give it continuity, and give it the importance and relevance that this research may have in the future.

Authors’ Response: Thanks for this recommendation: Please see lines 757-796 in the Discussion, part of which cover some aspects of potential follow-up work. Also, the following new paragraph has been added as the last paragraph in the Discussion section (lines 797-821 in the track change manuscript). “In addition to the above-mentioned activities, perhaps the most significant opportunity for future research identified is that offered by the establishment of strong, long-term relationships between community-based organizations (CBOs) that represent and reflect the voices of disadvantaged, underserved, and marginalized communities and academic institutions that are committed to community-based participatory research. CBOs can include independent community organizations, faith-based groups, health centers and community health workers, and many others. Working together, such partners can identify and address community needs, hazards, vulnerabilities, and capacities.  As demonstrated here and by Palinkas et al. [64] and Springgate et al. [65], CBOs such as LAMC not only may have a wealth of resources for partnering with researchers, but more importantly can connect researchers with community leaders and members, provide logistical and other support for community-based research and knowledge co-production, introduce academic researchers to locals who have significant traditional ecological and cultural information (i.e., “PhDs of the sidewalk”) [65], provide opportunities for student internships and research, and help legitimize the presence and activities of academics in their communities. Among many other things, academic researchers can connect local knowledge with new geospatial data and tools to assist communities in visualizing problems and issues and envisioning better futures.  Investments by academic institutions in long-term relationships with CBOs will provide a strong foundation for community-based academic research that can make enormous differences in development, acceptance, and implementation of useful and usable science to enhance community resilience in the face of global change.  Robust, long-term alliances between communities and academic institutions can create compelling forces for positive changes that last far beyond those accomplished via the usual project-by-project frameworks by which research is typically conducted.”

The last sentence in the Conclusions section was modified as well (see lines 841-843 in the track change revision). It now reads: “Bringing more people to the table to hear their opinions and concerns and creating vibrant, long-term partnerships between academic institutions and CBOs should be among the first steps towards more inclusive, equitable resilience-building”.

  1. The authors should add the limitations that this research had. What did the authors want to have done during the course of this research, and for whatever reason, it was not possible? What was or what were the difficulty/limitations that the authors felt and experienced? This information would be very useful to share, to help future research, and also help interpret some of the results obtained.

Authors’ Response: The section in the Discussion where limitations are noted has been revised and now reads as follows (see lines 736-756 in the track change manuscript): “An important limitation of this study is the relatively small sample size of community member interviews. While we attempted to schedule more interviews, Covid pandemic-related factors including substantially reduced ability to gather in person, reluctance of some non-community members to be interviewed, scheduling difficulties, and time available for the work restricted our ability to do so. Nevertheless, the richness of input received from participant interviews, augmented with information from participant-observation, as well as from secondary sources such as census records and spatial data, allowed this project to proceed in its aim of knowledge co-production. Another limitation of this study is that the majority of interview participants and field observations are representative of the Rosemont community. Nevertheless, slightly more than 40% of those interviewed were from other Charleston Neck communities. Since Rosemont falls in the City of Charleston and thus the USACE’s study area, it has received more attention and communication from both and therefore its residents are more engaged with and informed about the seawall project. Other communities are geographically close to Rosemont and they may be affected by the wall in similar ways, but at the time of the participant-observations, did not engage as much with this project. Also, all of the other Neck communities fall within the jurisdiction of the City of North Charleston which has not been involved significantly with the seawall project to date. While study results may be most applicable to Rosemont, all of the Neck communities were represented in spatial data collections, some had participants at one or more of the community meetings we observed, and all should be able to develop similar tools to leverage as needed based on our results.”

We thank the reviewer for his/her comments; they helped us improve the manuscript. Please see responses below and refer to the track-change revision of the manuscript.

Reviewer 2:

  1. Comments made directly in manuscript: Table 2:  affiliation lists 8 people in total while age rows have 7 only.

Fig. 13: Make sure the max value of color scale is same in both figures; A reference to vertical datum is also missing to properly interpret the data. Authors may add elevation above Mean sea level or NAVD88; and I-26 labeling is hiding DEM elevation, please remove or increase transparency.

Authors’ Response: The text has been modified to state that the city official was excluded from Table 2 to preserve anonymity and because different interview questions were used (see Table 1) (see lines 270-272 in the track change manuscript).  In addition, Table 2 has been retitled as “Community participant affiliation and demographic information” for clarity.

For Fig. 13, we have updated the figure, confirmed that the color scales are the same, added the vertical datum information to the figure heading, and removed the I-26 labels and added information about the highway location to the figure heading.

  1. The information in the Abstract of the paper is adequate and relevant, however authors should review the formatting of the Abstract as this is not the format of this journal. You should eliminate the topics: "Background", "Methods", "Results" and "Conclusions", and write the Abstract in a continuous form. The authors have followed the formatting of other journals, which is not correct. The authors should review this situation and correct it.

Authors’ Response: The Abstract has been modified as suggested. The last sentence in the abstract has been revised and now reads as follows: “Bringing more people to the table and creating vibrant, long-term partnerships between academic institutions and community-based organizations that include robust links to governmental organizations should be among the first steps in building inclusive, equitable, and climate resilient cities.”  See lines 38-40 in the track-change version of the manuscript.

  1. How was the questionnaire that was used in the interviews designed? How was this questionnaire initially validated? The authors state in the topic "2.2. Research Approach"(line 226-227) that the study was validated by the College of Charleston's Institutional Review Board (IRB), did this include, the initial validation of the questionnaire? This information should be clarified and- added to the article, to improve the scientific rigors of the article.  

Authors’ Response: This project did not use a survey methodology, but a community-based qualitative methodology that was triangulated between interviews and participant-observation. The interview question design was based on consultations with community leaders and through the direct participation of the LAMC organization, who ensured the relevance and validity of the qualitative data to collect. The text has been amended to reflect this (see lines 251-256 in track change revision).

  1. The authors' detailed explanation (line 244-251) of how the people who will participate in these interviews were approached is very well done, and leaves the scientific community in no doubt as to the validity of how they were conducted.

Authors’ Response: Thanks for your kind remarks and validation of our approach.

  1. The author indicates which sample was selected for this research, but does not identify which criteria he took into account for the selection of this sample. It would be very important for this information to be included in the article.

Authors’ Response: “Community members willing to contribute to this study were sampled via a snowball approach and identified through LAMC, word-of-mouth recommendations from community residents, and flyers distributed during community meetings and canvassing. Interviewees were chosen on the basis of their community affiliation, interest, and willingness to be interviewed. No attempt was made to select a “representative” sample of the populations as that was beyond the scope of the project and in accordance to community-based methodologies.” The text has been modified to clarify (see lines 257-263 in the track change manuscript).

  1. The author has not identified in the article, what is the Research Question, which led him to the development of this research. What does this article want to demonstrate? What was the gap that the author identified in his Literature Review (in other scientific articles), that this research intends to fill? This question is very important to be written in the article, to demonstrate the relevance and importance of this future publication to the scientific community.

Authors’ Response: The present study was undertaken to address the following question:  What, if any, concerns may environmental justice communities that are not included within the proposed sea wall have about how the wall and flooding might affect their communities?  We addressed this question by developing …”  (see lines 142-146 in the track change manuscript).

  1. How did the author carry out the analysis of the consistency of the data obtained? What is the reliability of the data obtained from the survey? This information is very relevant, and should be included in the article by the author.

Authors’ Response: The project did not use a survey methodology, but to ensure the reliability of the data, interview data were triangulated with the participant-observation data, and vice-versa, as well reviewed for reliability by the direct participation of the LAMC organization—all in accordance with typical qualitative and community-based methodologies.  The text has been revised to include this information (see lines 291-295 in the track change manuscript)

  1. The statistical analysis of the data is very weak and should be strongly improved to enhance the conclusions to be obtained. Why were statistical hypothesis tests not done? Has any significance level analysis of the data obtained been done? Collecting the answers and giving your result only, is not what is expected just in reading a scientific paper. This article is interesting, however, the author should review the answers and strongly strengthen his statistical analysis.

Authors’ Response: “Data collected in this study were from semi-structured interviews, not surveys, and participant observation during meetings, workshops, and other events in the communities. These are qualitative data that were not intended for standard statistical analysis for generalizability, because the researchers were searching for all possible community ideas regarding the cause and effects of flooding and plans for mitigation—an inductive rather than deductive research design. In addition, the interview sample size was insufficient to discern potential relationships. Our intent from the outset was to identify important themes via qualitative coding of interview responses and then to link them with spatial data using GIS, so statistical relevance of observed phenomena to map was not necessary. We employed a commonly used software, NVivo [33], to code qualitative responses to interview questions and notes taken during community meetings, following the approach of Cope [34]. (see lines 299-310 in the track change manuscript).

Additional explanation not included in the manuscript text: This inductive project aimed to generate theory from marginalized communities that claim not to be represented in the existing scientific literature. This community mistrust is why the research team chose to engage a knowledge co-production process and inductive research design. As such, this community-based methodology requires that researchers be open to exploring topics at the community direction, so the placement of hypotheses must be different than a survey research design. In a knowledge coproduction project, the intent is for the researchers to work with community members to generate hypotheses in the process of their work together, not at the outset of the study; the results section reviews what all these hypotheses are, and in one case the research team was able to further investigate a community observation via drone imagery. This kind of research process of community engagement is the main argument of the paper of how science working with the public ought to proceed.

  1. It would be very important that the authors at the end of the article indicate what future work can be developed from this research. It would be very good if this research did not stop here, given its potential, and that the scientific community could give it continuity, and give it the importance and relevance that this research may have in the future.

Authors’ Response: Thanks for this recommendation: Please see lines 757-796 in the Discussion, part of which cover some aspects of potential follow-up work. Also, the following new paragraph has been added as the last paragraph in the Discussion section (lines 797-821 in the track change manuscript). “In addition to the above-mentioned activities, perhaps the most significant opportunity for future research identified is that offered by the establishment of strong, long-term relationships between community-based organizations (CBOs) that represent and reflect the voices of disadvantaged, underserved, and marginalized communities and academic institutions that are committed to community-based participatory research. CBOs can include independent community organizations, faith-based groups, health centers and community health workers, and many others. Working together, such partners can identify and address community needs, hazards, vulnerabilities, and capacities.  As demonstrated here and by Palinkas et al. [64] and Springgate et al. [65], CBOs such as LAMC not only may have a wealth of resources for partnering with researchers, but more importantly can connect researchers with community leaders and members, provide logistical and other support for community-based research and knowledge co-production, introduce academic researchers to locals who have significant traditional ecological and cultural information (i.e., “PhDs of the sidewalk”) [65], provide opportunities for student internships and research, and help legitimize the presence and activities of academics in their communities. Among many other things, academic researchers can connect local knowledge with new geospatial data and tools to assist communities in visualizing problems and issues and envisioning better futures.  Investments by academic institutions in long-term relationships with CBOs will provide a strong foundation for community-based academic research that can make enormous differences in development, acceptance, and implementation of useful and usable science to enhance community resilience in the face of global change.  Robust, long-term alliances between communities and academic institutions can create compelling forces for positive changes that last far beyond those accomplished via the usual project-by-project frameworks by which research is typically conducted.”

The last sentence in the Conclusions section was modified as well (see lines 841-843 in the track change revision). It now reads: “Bringing more people to the table to hear their opinions and concerns and creating vibrant, long-term partnerships between academic institutions and CBOs should be among the first steps towards more inclusive, equitable resilience-building”.

  1. The authors should add the limitations that this research had. What did the authors want to have done during the course of this research, and for whatever reason, it was not possible? What was or what were the difficulty/limitations that the authors felt and experienced? This information would be very useful to share, to help future research, and also help interpret some of the results obtained.

Authors’ Response: The section in the Discussion where limitations are noted has been revised and now reads as follows (see lines 736-756 in the track change manuscript): “An important limitation of this study is the relatively small sample size of community member interviews. While we attempted to schedule more interviews, Covid pandemic-related factors including substantially reduced ability to gather in person, reluctance of some non-community members to be interviewed, scheduling difficulties, and time available for the work restricted our ability to do so. Nevertheless, the richness of input received from participant interviews, augmented with information from participant-observation, as well as from secondary sources such as census records and spatial data, allowed this project to proceed in its aim of knowledge co-production. Another limitation of this study is that the majority of interview participants and field observations are representative of the Rosemont community. Nevertheless, slightly more than 40% of those interviewed were from other Charleston Neck communities. Since Rosemont falls in the City of Charleston and thus the USACE’s study area, it has received more attention and communication from both and therefore its residents are more engaged with and informed about the seawall project. Other communities are geographically close to Rosemont and they may be affected by the wall in similar ways, but at the time of the participant-observations, did not engage as much with this project. Also, all of the other Neck communities fall within the jurisdiction of the City of North Charleston which has not been involved significantly with the seawall project to date. While study results may be most applicable to Rosemont, all of the Neck communities were represented in spatial data collections, some had participants at one or more of the community meetings we observed, and all should be able to develop similar tools to leverage as needed based on our results.”

We thank the reviewer for his/her comments; they helped us improve the manuscript. Please see responses below and refer to the track-change revision of the manuscript.

Reviewer 2:

  1. Comments made directly in manuscript: Table 2:  affiliation lists 8 people in total while age rows have 7 only.

Fig. 13: Make sure the max value of color scale is same in both figures; A reference to vertical datum is also missing to properly interpret the data. Authors may add elevation above Mean sea level or NAVD88; and I-26 labeling is hiding DEM elevation, please remove or increase transparency.

Authors’ Response: The text has been modified to state that the city official was excluded from Table 2 to preserve anonymity and because different interview questions were used (see Table 1) (see lines 270-272 in the track change manuscript).  In addition, Table 2 has been retitled as “Community participant affiliation and demographic information” for clarity.

For Fig. 13, we have updated the figure, confirmed that the color scales are the same, added the vertical datum information to the figure heading, and removed the I-26 labels and added information about the highway location to the figure heading.

  1. The information in the Abstract of the paper is adequate and relevant, however authors should review the formatting of the Abstract as this is not the format of this journal. You should eliminate the topics: "Background", "Methods", "Results" and "Conclusions", and write the Abstract in a continuous form. The authors have followed the formatting of other journals, which is not correct. The authors should review this situation and correct it.

Authors’ Response: The Abstract has been modified as suggested. The last sentence in the abstract has been revised and now reads as follows: “Bringing more people to the table and creating vibrant, long-term partnerships between academic institutions and community-based organizations that include robust links to governmental organizations should be among the first steps in building inclusive, equitable, and climate resilient cities.”  See lines 38-40 in the track-change version of the manuscript.

  1. How was the questionnaire that was used in the interviews designed? How was this questionnaire initially validated? The authors state in the topic "2.2. Research Approach"(line 226-227) that the study was validated by the College of Charleston's Institutional Review Board (IRB), did this include, the initial validation of the questionnaire? This information should be clarified and- added to the article, to improve the scientific rigors of the article.  

Authors’ Response: This project did not use a survey methodology, but a community-based qualitative methodology that was triangulated between interviews and participant-observation. The interview question design was based on consultations with community leaders and through the direct participation of the LAMC organization, who ensured the relevance and validity of the qualitative data to collect. The text has been amended to reflect this (see lines 251-256 in track change revision).

  1. The authors' detailed explanation (line 244-251) of how the people who will participate in these interviews were approached is very well done, and leaves the scientific community in no doubt as to the validity of how they were conducted.

Authors’ Response: Thanks for your kind remarks and validation of our approach.

  1. The author indicates which sample was selected for this research, but does not identify which criteria he took into account for the selection of this sample. It would be very important for this information to be included in the article.

Authors’ Response: “Community members willing to contribute to this study were sampled via a snowball approach and identified through LAMC, word-of-mouth recommendations from community residents, and flyers distributed during community meetings and canvassing. Interviewees were chosen on the basis of their community affiliation, interest, and willingness to be interviewed. No attempt was made to select a “representative” sample of the populations as that was beyond the scope of the project and in accordance to community-based methodologies.” The text has been modified to clarify (see lines 257-263 in the track change manuscript).

  1. The author has not identified in the article, what is the Research Question, which led him to the development of this research. What does this article want to demonstrate? What was the gap that the author identified in his Literature Review (in other scientific articles), that this research intends to fill? This question is very important to be written in the article, to demonstrate the relevance and importance of this future publication to the scientific community.

Authors’ Response: The present study was undertaken to address the following question:  What, if any, concerns may environmental justice communities that are not included within the proposed sea wall have about how the wall and flooding might affect their communities?  We addressed this question by developing …”  (see lines 142-146 in the track change manuscript).

  1. How did the author carry out the analysis of the consistency of the data obtained? What is the reliability of the data obtained from the survey? This information is very relevant, and should be included in the article by the author.

Authors’ Response: The project did not use a survey methodology, but to ensure the reliability of the data, interview data were triangulated with the participant-observation data, and vice-versa, as well reviewed for reliability by the direct participation of the LAMC organization—all in accordance with typical qualitative and community-based methodologies.  The text has been revised to include this information (see lines 291-295 in the track change manuscript)

  1. The statistical analysis of the data is very weak and should be strongly improved to enhance the conclusions to be obtained. Why were statistical hypothesis tests not done? Has any significance level analysis of the data obtained been done? Collecting the answers and giving your result only, is not what is expected just in reading a scientific paper. This article is interesting, however, the author should review the answers and strongly strengthen his statistical analysis.

Authors’ Response: “Data collected in this study were from semi-structured interviews, not surveys, and participant observation during meetings, workshops, and other events in the communities. These are qualitative data that were not intended for standard statistical analysis for generalizability, because the researchers were searching for all possible community ideas regarding the cause and effects of flooding and plans for mitigation—an inductive rather than deductive research design. In addition, the interview sample size was insufficient to discern potential relationships. Our intent from the outset was to identify important themes via qualitative coding of interview responses and then to link them with spatial data using GIS, so statistical relevance of observed phenomena to map was not necessary. We employed a commonly used software, NVivo [33], to code qualitative responses to interview questions and notes taken during community meetings, following the approach of Cope [34]. (see lines 299-310 in the track change manuscript).

Additional explanation not included in the manuscript text: This inductive project aimed to generate theory from marginalized communities that claim not to be represented in the existing scientific literature. This community mistrust is why the research team chose to engage a knowledge co-production process and inductive research design. As such, this community-based methodology requires that researchers be open to exploring topics at the community direction, so the placement of hypotheses must be different than a survey research design. In a knowledge coproduction project, the intent is for the researchers to work with community members to generate hypotheses in the process of their work together, not at the outset of the study; the results section reviews what all these hypotheses are, and in one case the research team was able to further investigate a community observation via drone imagery. This kind of research process of community engagement is the main argument of the paper of how science working with the public ought to proceed.

  1. It would be very important that the authors at the end of the article indicate what future work can be developed from this research. It would be very good if this research did not stop here, given its potential, and that the scientific community could give it continuity, and give it the importance and relevance that this research may have in the future.

Authors’ Response: Thanks for this recommendation: Please see lines 757-796 in the Discussion, part of which cover some aspects of potential follow-up work. Also, the following new paragraph has been added as the last paragraph in the Discussion section (lines 797-821 in the track change manuscript). “In addition to the above-mentioned activities, perhaps the most significant opportunity for future research identified is that offered by the establishment of strong, long-term relationships between community-based organizations (CBOs) that represent and reflect the voices of disadvantaged, underserved, and marginalized communities and academic institutions that are committed to community-based participatory research. CBOs can include independent community organizations, faith-based groups, health centers and community health workers, and many others. Working together, such partners can identify and address community needs, hazards, vulnerabilities, and capacities.  As demonstrated here and by Palinkas et al. [64] and Springgate et al. [65], CBOs such as LAMC not only may have a wealth of resources for partnering with researchers, but more importantly can connect researchers with community leaders and members, provide logistical and other support for community-based research and knowledge co-production, introduce academic researchers to locals who have significant traditional ecological and cultural information (i.e., “PhDs of the sidewalk”) [65], provide opportunities for student internships and research, and help legitimize the presence and activities of academics in their communities. Among many other things, academic researchers can connect local knowledge with new geospatial data and tools to assist communities in visualizing problems and issues and envisioning better futures.  Investments by academic institutions in long-term relationships with CBOs will provide a strong foundation for community-based academic research that can make enormous differences in development, acceptance, and implementation of useful and usable science to enhance community resilience in the face of global change.  Robust, long-term alliances between communities and academic institutions can create compelling forces for positive changes that last far beyond those accomplished via the usual project-by-project frameworks by which research is typically conducted.”

The last sentence in the Conclusions section was modified as well (see lines 841-843 in the track change revision). It now reads: “Bringing more people to the table to hear their opinions and concerns and creating vibrant, long-term partnerships between academic institutions and CBOs should be among the first steps towards more inclusive, equitable resilience-building”.

  1. The authors should add the limitations that this research had. What did the authors want to have done during the course of this research, and for whatever reason, it was not possible? What was or what were the difficulty/limitations that the authors felt and experienced? This information would be very useful to share, to help future research, and also help interpret some of the results obtained.

Authors’ Response: The section in the Discussion where limitations are noted has been revised and now reads as follows (see lines 736-756 in the track change manuscript): “An important limitation of this study is the relatively small sample size of community member interviews. While we attempted to schedule more interviews, Covid pandemic-related factors including substantially reduced ability to gather in person, reluctance of some non-community members to be interviewed, scheduling difficulties, and time available for the work restricted our ability to do so. Nevertheless, the richness of input received from participant interviews, augmented with information from participant-observation, as well as from secondary sources such as census records and spatial data, allowed this project to proceed in its aim of knowledge co-production. Another limitation of this study is that the majority of interview participants and field observations are representative of the Rosemont community. Nevertheless, slightly more than 40% of those interviewed were from other Charleston Neck communities. Since Rosemont falls in the City of Charleston and thus the USACE’s study area, it has received more attention and communication from both and therefore its residents are more engaged with and informed about the seawall project. Other communities are geographically close to Rosemont and they may be affected by the wall in similar ways, but at the time of the participant-observations, did not engage as much with this project. Also, all of the other Neck communities fall within the jurisdiction of the City of North Charleston which has not been involved significantly with the seawall project to date. While study results may be most applicable to Rosemont, all of the Neck communities were represented in spatial data collections, some had participants at one or more of the community meetings we observed, and all should be able to develop similar tools to leverage as needed based on our results.”

Round 2

Reviewer 2 Report

Congratulations to the authors, for the work of revision and improvement they have done to the article!